

# Dating and morphostratigraphy of uplifted marine terraces in the Makran subduction zone (Iran)

Raphaël Normand[1], Guy Simpson[1], Frédéric Herman[2], Rabiul Haque Biswas[2], Abbas Bahroudi[3] & Bastian Schneider[4]

[1]Department of Earth Sciences, University of Geneva, Rue des Maraichers 13, 1205 Genève, Switzerland.
[2]Institute of Earth Surface Dynamics, Faculty of Geosciences and Environment, University of Lausanne, 1012 Lausanne, Switzerland.
[3]Exploration department, School of Mining Engineering, University of Tehran, Northern Kargar avn, P.O. Box 11365-4563, Tehran, Iran.
[4]Steinmann Institute of Geology, University of Bonn, Nussallee 8, 53115 Bonn, Germany.

*Correspondence to*: Rapahël Normand (Raphael.Normand@unige.ch)

**Abstract.** The western part of the Makran subduction zone (Iran) has not experienced a great megathrust earthquake in recent human history, yet, the presence of emerged marine terraces along the coast indicates that the margin has been tectonically active during at least the late Quaternary. To better understand the surface deformation of this region, we mapped the terraces sequences of seven localities along the Iranian Makran. Additionnaly, we performed radiocarbon, $^{230}$Th/U and optically stimulated luminescence (OSL) dating of the layers of marine sediments deposited on top of the terraces. This enabled us to correlate the terraces regionally and to assign them to different Quaternary sea level highstands. Our results show east-west variations in surface uplift rates mostly between 0.05 and 1.2 mm y$^{-1}$. We detected a region of anomalously high uplift rate, where two MIS 3 terraces are emerged, yet we are uncertain how to insert these results in a geologically coherent context. Although it is presently not clear whether the uplift of the terraces is linked with the occurrence of large megathrust earthquakes, our results highlight heterogeneous accumulation of deformation in the overriding plate.



# 1. Introduction

Surface uplift at the coastline of convergent margins is mainly the result of long term accumulation of coseismic and interseismic deformation (King et al., 1988; Lajoie, 1986; Matsu'ura and Sato, 1989; Segall, 2010; Simpson, 2015; Wesson et al., 2015). At most subduction zones around the globe, Quaternary uplift is expressed by the occurrence of sequences of emerged marine terraces (Anderson et al., 1999; Burbank and Anderson, 2001; Henry et al., 2014; Keller and Pinter, 2002; Lajoie, 1986; Pedoja et al., 2014; Pirazzoli, 1994; Trenhaile, 2002). A key aspect of the study of marine terrace sequences is the dating of the geomorphic surfaces and/or deposits of the successive terraces. These types of studies provide a morphostratigraphic framework for the interpretation of the terraces (i.e. correlation to past sea-level highstands) and ultimately they enable calculation of surface uplift rates (Jara-Muñoz et al., 2015; Lajoie, 1986; Meschis et al., 2018; Pedoja et al., 2018a; Regard et al., 2017; Roberts et al., 2013; Saillard et al., 2011).

The Makran subduction zone (MSZ) of southern Iran and Pakistan hosts spectacular examples of recently uplifted marine terraces. These were described as early as the 1940s (Falcon, 1947; Harrison, 1941), but have remained largely unstudied in the last few decades. Most dating attempts in the Makran were performed by [14]C dating, which has the inconvenience of being uncertain beyond ca. 20 ka (Murray-Wallace and Woodroffe, 2014). Therefore, attribution of the different Makran terrace levels to past sea-level highstands has not yet been attempted. Indeed, amongst the subduction zones hosting marine terraces, the MSZ is one of the few cases where knowledge of the altitude of the global geodynamic marker, the Marine Isotopic Stage (MIS) 5e benchmark, is still lacking (Pedoja et al., 2014).

The long-term seismic behavior in the MSZ remains largely enigmatic due to the poor historical seismic data and the low recorded seismicity since installation of modern seismometers (1964) (Zare et al., 2014). Although the eastern Pakistan segment of the MSZ has experienced recent large thrust earthquakes (Byrne et al., 1992; Penney et al., 2017), the western Iranian segment, the focus of this study, seems to have been aseismic for more than 500-100 years and potentially has been accumulating strain since then (Byrne et al., 1992; Penney et al., 2017; Rajendran et al., 2013). Because the Makran region is sparsely populated, local risks associated with earthquakes are moderate. However, a tsunami induced by an earthquake of the MSZ could reach densely populated areas such as Muscat, Karachi or the western coast of India (Hafeez, 2007; Heidarzadeh and Kijko, 2011; Schneider et al., 2016). Emerged sequences of marine terraces record the long-term uplift trend averaged over multiple seismic cycles and, in that regard, give valuable information on the tectonic activity of the margin (Lajoie, 1986). Here, we investigate ten sequences of marine terraces along the western Makran using modern dating techniques combined with remote sensing. We present seven terrace maps, resulting from extensive fieldwork, with the help of satellite imagery and a 0.4 arcsec (~12 m) resolution DEM. We dated 13 radiocarbon, 3 [230]Th/U and 12 optically stimulated luminescence (OSL) samples of the marine sedimentary layer covering the lower (younger) terraces in order to correlate the dated terraces of the sequences to past sea-level highstands, calculate uplift rates and based on these rates, estimate the age of the older mapped





terraces. We were then able to identify the MIS 5e benchmark along the 200 km long studied segment of the MSZ, improving the database on Cenozoic sequences of paleoshorelines (Pedoja et al., 2014). Moreover, we observe uplift variations along strike the subducting trench, possibly linked to coupling of the subduction interface (e.g.: Jara-Muñoz et al., 2015; Saillard et al., 2017), increasing our knowledge of the seismic behavior of the MSZ.

## 5  2. Settings

### 2.1 Geodynamics and seismicity of the Makran subduction zone (MSZ)

Our study is focused on the coastal plain of the MSZ, southeast Iran (Fig. 1). This coastal strip is located near the middle of a large active accretionary complex, part of which is offshore, and part of which lies behind the studied area further to the north (White and Ross, 1979). The Makran trench strikes roughly in an east-west direction and the slab is dipping to the north at a

shallow angle of up to 3° (Kopp et al., 2000; Manaman et al., 2011; White and Louden, 1982). The record from GPS surveys around the Oman Sea indicates active convergence between the subducting Arabian and overriding Eurasian plates of about 2 cm y$^{-1}$ (Frohling and Szeliga, 2016; Khan et al., 2008; Masson et al., 2007; Vernant et al., 2004). Although most of this convergence seems to be accommodated by reverse faults within the prism, margin-parallel normal faulting is predominant in the coastal region (Back and Morley, 2016; Burg et al., 2012; Ghorashi, 1978; Harms et al., 1984; Hosseini-Barzi and Talbot,

2003; Little, 1972; Platt and Leggett, 1986; Snead, 1993).

The seismic activity of the MSZ is low compared to other subduction zones around the globe, although most studies agree that the potential for tsunamigenic megathrust earthquakes exists (e.g.: Heidarzadeh et al., 2008; Hoffmann et al., 2013a; Pararas-Carayannis, 2006; Penney et al., 2017; Shah-Hosseini et al., 2011; Smith et al., 2013). The Middle East earthquake catalog from Zare et al., 2014, ranging from 1220 BC to 2006, shows that earthquakes of Mw >5 are less densely clustered in the MSZ

than in other seismically active regions of the Middle East, like the Zagros or Caucasus. Byrne et al., 1992 pointed out that the eastern and the western segment of the MSZ seem to have a different seismic behavior. The eastern part is more active, with several thrust earthquakes occurring in the region of Pasni, notably the Mw 8.1 event of 1945 (Byrne et al., 1992; Hoffmann et al., 2013b; Quittmeyer and Jacob, 1979) and a recent event of Mw 6.3 in 2017 (Penney et al., 2017). In the western Makran, the two last great earthquakes are historical events from 1008 and 1483 (Ambraseys and Melville, 1982; Heidarzadeh et al.,

2008), though their exact magnitude, position and focal mechanism are poorly known (Musson, 2009). The sinistral strike-slip Sonne Fault that obliquely crosses the wedge off Gwadar city has been proposed as the limit between the eastern and western segments of the MSZ (Kukowski et al., 2000; Rajendran et al., 2013), for which different tomographical properties have been identified (Al-Lazki et al., 2014; Manaman et al., 2011). Although the western MSZ is seismically quiet, evidence of past earthquakes, activity in the eastern segment and ongoing plate convergence indicate that the plate boundary might have been

accumulating elastic strain for more than 500 years. Thus, the western MSZ has the potential to produce a catastrophic earthquake (Penney et al., 2017; Rajendran et al., 2013; Smith et al., 2013).



## 2.2 Geological and geomorphological setting

Along the western Makran coast, the bedrock lithology is dominated by highly erodible Neogene-recent fine-grained slope sedimentary rocks, forming a wide flat coastal plain (Blanford, 1872; Burg et al., 2012; Falcon, 1947; Ghorashi, 1978; Harms et al., 1984; Harrison, 1941; McCall, 2002; Page et al., 1979; Snead, 1967; Stiffe, 1874). The coastal plain is locally interrupted

and overlooked by prominent rocky headlands, 10 to 30 km in length, hosting emerged sequences of marine terraces. The lithology of these headlands alternates between coarse coastal and fine slope deposits, similar to those found in the Makran Ranges, northwards of the coastal plain (Harms et al., 1984). The headlands are separated by wide omega-shaped bays that exhibit sequences of Holocene prograding beach ridges up to 10 km long (Gharibreza, 2016; Shah-Hosseini et al., 2018) (Fig. 1).

The Makran coast is facing the Oman Sea, situated in the northern part of the Indian ocean, and has a tide range between 1.8 and 4 meters (Sanlaville et al., 1991; Snead, 1993). The studied area of Chabahar receives waves coming mostly from the SSE and SW, with significant wave height of 1 to 3 m and periods between 4 and 8 s (Saket and Etemad-shahidi, 2012). The climate is arid to semi-arid and vegetation is sparse. Precipitation rates remain low (mean annual precipitation: ~120 mm), which implies low erosion rates on sandstone rocks (Haghipour et al., 2012; Page et al., 1979), although fine-grained bedrock gets

strongly eroded during the yearly rainfall events (Falcon, 1947; Ghorashi, 1978; Harrison, 1941; Snead, 1967; Stiffe, 1874).

## 2.3 Previous work on the sequences of Makran marine terraces

The uplifting nature of the MSZ coastline has been recognized as early as the 1940s (Falcon, 1947; Harrison, 1941). Since then, the Pleistocene marine terraces have been the focus of several studies (e.g.: Little, 1972; Page et al., 1979; Reyss et al., 1998; Snead, 1967, 1993, Vita-Finzi, 1975, 1980, 1981, 1982, 2002). The marine terraces of the Makran have been described

as flat topped, isolated hills, with a wave-cut unconformity capped by a shelly sandstone layer of shoreface to foreshore deposits (Falcon, 1947; Little, 1972; Page et al., 1979; Snead, 1993). This layer of marine sediment is an important characteristic of the Makran marine terraces and will hereafter be referred to as "terrace deposits". Since the marine terraces are built on top of the Tertiary sedimentary sequences of the Makran accretionary prism, the bedrock sandstones units can be misinterpreted as terrace deposits. Snead (1993) has defined the term "structural terrace" which refers to those subhorizontal

Tertiary sandstone beds, resistant to erosion, which are not necessarily formed by wave-erosion and therefore do not correspond to a paleo sea-level altitude. Maps of the terraces have been published by a few authors (Little, 1972; Page et al., 1979; Snead, 1993) and give valuable spatial information on their extent. Some sequences of terraces have been described (Page et al., 1979; Reyss et al., 1998), although the linking of the successive marine terraces of a sequence to former sea-level highstands (Lajoie, 1986) has not yet been attempted.

Dating of the MSZ marine terrace deposits has been undertaken by several authors (Haghipour et al., 2014; Little, 1972; Page et al., 1979; Rajendran et al., 2013; Reyss et al., 1998; Vita-Finzi, 1975, 1980, 1981) in order to infer surface uplift rates. Due to the lack of coral reefs in the MSZ, most of these results come from [14]C dating of mollusk shells. Ages are either the second





half of the Holocene (< 6 ka) or older than 20 ka. Some authors have argued that results older than 20 ka should be minimum ages, due to the limitations of the $^{14}$C method (Haghipour et al., 2014; Page et al., 1979; Rajendran et al., 2013; Reyss et al., 1998; Vita-Finzi, 1975, 1980, 1981). The only dated terraces older than the last glacial period are from Page et al. (1979), who dated mollusk shells with a combination of $^{230}$Th/$^{234}$U and $^{231}$Pa/$^{235}$U methods. They dated the Jask terrace and one level of the

Konarak terraces with these methods, both yielding an age coeval within errors to the last maximum interglacial (MIS 5e). These ages are considered reliable by the authors. We compiled a summary of previously published ages of the Makran terraces (and beaches) in supplementary data Table A.

## 2.4 Pleistocene sea-level curve

A key aspect of the study of uplifted marine terraces is a good knowledge of past variations in eustatic sea-level. Numerous

works aiming at reconstructing eustatic sea-level curves for the Late Quaternary have been performed. While the general trends are known, uncertainties concerning the timing and magnitude of sea-level fluctuations are still significant. The most popular proxies used to reconstruct eustatic sea-level curves are the $\delta^{18}$O record of benthic foraminifera (e.g.: Arz et al., 2007; Rohling et al., 2014, 2009, Siddall et al., 2006, 2003; Waelbroeck et al., 2002) and dating of constructional coral reef terraces on uplifted coasts (e.g.: Bard et al., 1990; Bloom et al., 1974; Chappell, 2002; Chappell et al., 1996; Cutler et al., 2003; Esat and

Yokoyama, 2006; Fairbanks, 1989; Hibbert et al., 2016; Medina-Elizalde, 2013; Yokoyama et al., 2001). Coral-based sea-level reconstructions are sensitive to global uncertainties such as coral dating precision and coral growth depth distribution, as well as uncertainties inherent to each locality, such as uplift rate variations and glacial-isostatic adjustment (GIA) (Creveling et al., 2017; Hibbert et al., 2016; Medina-Elizalde, 2013; Murray-Wallace and Woodroffe, 2014). $\delta^{18}$O records are more continuous through time, but their accuracy are also limited by dating and measurement errors, local variations in sea-water

temperature and salinity, and uncertainties regarding the relation between the $\delta^{18}$O record, ice volume and eustatic sea-level (in magnitude as well as in timing) (Spratt and Lisiecki, 2016; Waelbroeck et al., 2002). Because of these uncertainties, many different eustatic sea level curves have been published and it is not straightforward to choose any one curve as a reference.

## 3. Methods

### 3.1 Mapping

Three field campaigns of one month each were performed in the coastal area of the Iranian Makran. Outcrops from Jask to the west until Pasabander in the east were visited (Fig. 1). Spatial information was also gained from the study of satellite images (sparse vegetation cover) and from the TanDEM-X DEM (0.4 arcsec / ~12 m resolution and 2 m vertical error). Terrace maps were digitalized using the ToolMap 2.6.2035 (MEYRIN) software.

Throughout the field campaigns, we used a series of criteria that helped to differentiate actual marine terraces from other flat

surfaces, such as structural terraces. These criteria were determined from observation of marine terraces as well as their modern equivalent along the coast of the Iranian Makran (Fig. 2) (supplementary data B.1). Not all criteria were met in each mapped





terrace due to variability in factors such as bedrock lithology, erosion and anthropogenic disturbances. Terrace levels were attributed to a MIS based on our dating results. Undated terraces have been assigned to successive highstands based on their shoreline angle altitude profile. We tried to fit them best to a constant uplift scenario, keeping in mind that some terraces of the sequence might have been completely eroded. Moreover, attribution of terraces to a correct MIS was also complicated by

the action of normal faults. To illustrate the reliability of our terrace maps, we added confidence indexes to our terrace boundaries and our MIS attribution (1 = low confidence, 5 = high confidence). This information is available within the GIS files of the terrace maps (Normand et al., 2018).

## 3.2 Dating

### 3.2.1 Dating target

The dating techniques used in this paper focus on marine materials deposited above the wave-cut platform of the marine terraces (Fig. 2a, 2c, 2i). Therefore, the time relationship between the platform formation by wave erosion (i.e. our sea-level reference for uplift calculation) and the deposition of sediments (OSL dating) or shells (radiometric dating) above this erosive surface needs to be clarified. We studied the stratigraphy of the terrace deposits, whose logs are reported in Fig. 4i, 5j, 6e and 7b. All studied logs show a similar general trend consisting of a wave cut surface overlain by a prograding sequence of

shoreface to foreshore facies (fig. 2a, 2c). Platform carving usually happens at the beginning of the highstand, until the platform gets too wide and the wave energy too dissipated to carry out effective cliff erosion (Anderson et al., 1999; Trenhaile, 2000). The occurrence of ravinement deposits at the base of the terrace deposits also supports the idea of platform carving during the transgressive event (Fig. 2c) (Catuneanu et al., 2011). We interpret the prograding terrace deposits to be deposited on the platform after the erosive period; during the end of the sea-level stillstand and the start of the sea-level fall (e.g.: Jara-Muñoz

and Melnick, 2015). Hence, we believe that terrace carving and sediment deposition both occur within the same highstand. This is true for the Makran Holocene beaches, which were deposited since the mid-Holocene highstand (Gharibreza, 2016; Gharibreza and Motamed, 2006; Sanlaville et al., 1991; Shah-Hosseini et al., 2018).

### 3.2.2 $^{14}$C dating

Shell samples were collected from the terrace deposits. We analyzed mostly calcitic mollusk shells (like oysters), but also three

aragonitic mollusks. Samples were sent to Beta Analytics Inc. where they were prepared, bleached and analyzed with the traditional AMS counting method. In parallel, samples were observed with the SEM and analyzed with X-ray diffraction (XRD) to estimate their state of recrystallization. XRD is a useful tool for this task since an aragonitic shell should not contain calcite unless it is recrystallized. Unfortunately, we could not apply this method to calcitic shells. $^{14}$C ages were calibrated to calendar ages using Oxcal 4.2 (Bronk Ramsey and Lee, 2013), with the calibration curves IntCal 13 and Marine 13 (Reimer

et al., 2013), and a delta_R value of 236±31, as calculated from the website http://calib.org/marine/ based on the local values





of von Rad et al. (1999) and Southon et al. (2002). Dating results, analytical information, XRD results and SEM images of shells are provided in Table 1 and Normand et al. (2018).

### 3.2.3 $^{230}$Th/U dating

Since most $^{14}$C ages are close to the limit of the method (20-50 ky), we performed $^{230}$Th/U dating on the three aragonitic mollusk shells previoulsy dated by $^{14}$C to test the validity of the $^{14}$C results. $^{230}$Th/U dating on mollusk shells is uncommon because the Th/U system is known to be easily re-opened during diagenesis (Hillaire-Marcel et al., 1996; Kaufman et al., 1971). To account for this, $^{234}$U/$^{238}$U activity ratios were also measured. If this ratio is close to that of seawater ($1.14 \pm 0.014$, Arslanov et al., 2002), the system is interpreted to have been closed shortly after shell crystallization and the age can be interpreted as true. However, if the ratio is higher, then the system was probably contaminated by Uranium from the groundwater and the measured age can only be interpreted as a minimum age (Arslanov et al., 2002; Causse et al., 2003; Hillaire-Marcel et al., 1996; Kaufman et al., 1996). Samples were analyzed in the Geotop lab of radiochronology at the University of Montreal. Analytical information are provided in Table 1 and Normand et al. (2018).

### 3.2.4 OSL dating

We sampled rocks of foreshore and shoreface facies from the terrace deposits, which are known to be good candidates for OSL dating because sediments deposited in such conditions have a high chance of complete bleaching before burial (Lamothe, 2016; Murray and Olley, 2002). Samples are 10-15 cm sized rock blocks, from which we extracted the core under red dim light. We treated the samples with the usual preparation methods to isolate 90-150 µm feldspar and quartz grains through sequential treatment with HCl and $H_2O_2$, sodium polytangstate density separation, Frantz magnetic separation and HF treatment to quartz grains. Although both the minerals, quartz and feldspar, were separated, we ended up rejecting quartz due to their OSL signal being contaminated by feldspar inclusion (e.g.: Lawson et al., 2015).

To determine the burial dose or equivalent dose ($D_e$), we measured the luminescence signals on 12 aliquots (each containing ~100 grain) per sample using the Risø TL/OSL-DA-20 reader at the Institute of Earth Surface Dynamics, University of Lausanne. To minimize the effect of anomalous fading (Wintle, 1973), commonly observed in IRSL of feldspar, we measured the elevated-temperature (225°C) post-IR IRSL SAR protocol (Buylaert et al., 2009). Results were processed with the Analyst 4.31.7 software (Duller, 2015). Each aliquot was evaluated according to the following acceptance criteria: recycling ratios at 10%, maximum test dose error at 10%, maximum recuperation at 10% of the natural signal and maximum paleodose error at 20%. $D_e$ values were assessed using the central age model (Galbraith et al., 1999). The environmental dose was calculated with the DRAC software (Durcan et al., 2015) after measuring the radioactive elements (U, Th, K and Rb) using ICPMS (from ActLabs, Canada). The reliability of the protocol and zeroing of clock at the time of deposition was assessed with a dose-recovery test (Murray and Wintle, 2003; Wallinga et al., 2000) on 4 representative samples. We exposed the samples to natural light for 48 continuous hours before measuring the natural signals (to check the residual dose) and recovery of an artificially



given dose of 300 s (~36 Gy) on the top of artificially bleached sample using the same pIR-IRSL protocol. Dose recovery ratios (recovered dose/ given dose) are 0.9 to 1.1.

Although the post-IR IRSL of feldspar at elevated temperature is less prone to anomalous fading (Buylaert et al., 2009; Thomsen et al., 2008), we performed a fading test on all samples to correct this effect. We measured 4 aliquots per sample

following the procedure of Buylaert et al. (2009) to determine the fading rate (g-values; (Auclair et al., 2003)) of each sample. Ages were then corrected accordingly, using Eq. (A5) of Huntley and Lamothe (2001), except for two samples (RN17-14 and RN17-24), which did not show any fading behavior. Although this fading correction method appropriate to younger sample, where equivalent dose is in the linear part of dose response curve, we applied this method to all samples (some are in the non linear part of dose response curve), considering that the fading rate of pIR-IRSL225 signal is generally small.

Results are presented in Table 2. More details on sample measurements, environmental dose and age calculations are presented in Normand et al. (2018).

## 3.3 Calculation of surface uplift rate

The sequence of uplifted marine terraces observed in the landscape is assumed to be the geomorphic record of Quaternary sea-level highstands (Lajoie, 1986) which happened during odd numbered MIS. In general, the age and altitude of the sampled

material is not used to calculate uplift rates. Instead, the age of the nearest sea-level highstand and the elevation of the shoreline angle (also called inner edge, or inner margin) (Fig. 2a) of the terrace is used (e.g.: Jara-Muñoz et al., 2015; Pedoja et al., 2018a, 2018b; Roberts et al., 2013; Saillard et al., 2009). Uplift rates (U) are then calculated using the terrace shoreline angle elevation (E), eustatic sea-level estimates (e) and age (A) of the correlated highstand (from the literature), with the formula (Fig. 3) (Lajoie, 1986):

$U = (E\text{-}e) / A$ (1)

Uplift rates may vary over time, whereas Eq.(1) considers only the mean uplift rate from today until the age A. To get more insight on uplift variations though time, different terrace levels within the same profile must be investigated and their individual uplift rates compared (e.g.: Saillard et al., 2009).

Although this equation is simple, it is difficult to get precise uplift rates because of the errors on the different terms, especially

on the eustatic sea-level curve (e) (see Sect. 2.4). Therefore, we calculated uplift rate ranges to include the different uncertainties using a modified version of Eq.(1) (e.g.: Pedoja et al., 2018a, 2018b)(Pedoja et al., 2018b, 2018a) (Fig. 3);

$U_{min} = [(E - \Delta E) - (e + \Delta e)] / (A + \Delta A)$ (2)

$U_{max} = [(E + \Delta E) - (e - \Delta e)] / (A - \Delta A)$ (3)

where the delta symbols represent the estimated variability in the different parameters. In an age vs height diagram, $U_{min}$

represent the curve with the lowest slope and $U_{max}$ that with the steepest (Fig. 3). U is the mean uplift rate, as calculated using Eq.(1). We also computed apparent uplift rates ($U_a$), calculated without any eustatic correction (e.g.: Pedoja et al., 2018b), which is a means of comparing terraces globally, without considering the choice of the eustatic component (Table 3).





### 3.3.1 Past highstand age and eustatic sea-level

In this paper, we calculate uplift rates based on the eustatic curves from two recent papers, namely Shakun et al. (2015) and Spratt and Lisiecki (2016) (sea-level estimates used are reported in supplementary data Table B.2). We chose these curves because they are built from a multitude of different $\delta^{18}O$ records from all around the globe, using statistical tools, and therefore

we feel that they could potentially be a good approximation of global sea-level change. Moreover, they also provide error ranges, which are useful to calculate uplift rate errors.

Due to the age uncertainties of the $\delta^{18}O$ records, the timing of the chosen sea-level curves is not well constrained (Shakun et al., 2015; Spratt and Lisiecki, 2016). We use the MIS numbering proposed by Railsback et al., 2015 and MIS ages compiled from the literature as reference (Dutton et al., 2009; Murray-Wallace and Woodroffe, 2014; Stirling et al., 1998, 2001)

(supplementary data Table B.2). If sea-level reconstructions get more precise in the future, it is always possible and easy to modify the ages and eustatic values and re-calculate the uplift rates.

### 3.3.2 Shoreline angle

Although it is not perfect as zero sea-level indicator (Rovere et al., 2016), the shoreline angle of each terrace (see Fig. 2a) is a good approximation of sea-level during the peak of a highstand (Burbank and Anderson, 2001; Lajoie, 1986). In the Makran,

precise measurement of the shoreline angle altitude is complicated because:

(1) Most of the time it is buried below a sedimentary layer (Fig. 2a, 2i).

(2) A peculiar morphological characteristic of the Makran terraces is that the paleocliff at the back of the terrace is sometimes eroded (Fig. 2h). Therefore, the exact position and especially the altitude of the shoreline angle is unknown, since marine terraces, as their modern counterparts (rocky shore platforms), generally slope at a shallow angle towards the sea. In this case,

the wave-cut surface at the back of the terrace was used as an approximation for the shoreline angle. This type of shoreline angle will hereafter be referred to as "eroded shoreline angle" and uplift rates calculated from their altitude might underestimate the reality.

(3) Most terraces are tilted parallel to the subduction trench, therefore the altitude of the shoreline angle (and the uplift rate) varies from east to west. We calculated uplift rates for each sample using the terrace shoreline angle situated directly

northwards of the sample (Table 3) (i.e. perpendicular to the trench), from which we subtracted the sediment thickness observed on the field to get our $E \pm \Delta E$ values (Fig. 3) (supplementary data Table B.3).

Since the terraces are tilted, measuring uplift rates at a point is only relevant locally. To illustrate regional terrace tilting, we calculated lateral variation in uplift rate using the shoreline angle altitude extracted from the DEM (Fig. 4b, 4g, 5b, 5e, 5h, 6b, 7e) (DEM vertical error, $\Delta Alti = 2$ m) from which we subtracted a general value for the thickness of the sediments. Based on

our field observations of the Makran terraces, the sediment layer can vary from 2 to 10 meters, so we used $S \pm \Delta S = 6 \pm 4$ m. We chose the sea-level curve of Spratt and Lisiecki (2016) for a eustatic reference (supplementary data Table B.2) and calculated uplift rates and their errors following the method described previously (Sect. 3.3) (Fig. 4c, 4h, 5c, 5f, 5i, 6c, 7f).



# 4. Results

## 4.1 Mapping

We present here seven maps of the terrace regions (from east to west: Jask, Tang, Gurdim, Koanrak, Chabahar - Ramin, Lipar, Beris and Pasabander) (Fig. 4-7). Map legends are found in Fig. 4e. The maps are accompanied by topographical,

sedimentological and geodynamical data which are: 1) The reports of selected north-south topographic profiles through the terraces. 2) A report of the shoreline angle altitude per latitude value (i.e. alongstrike the subducting trench). 3) A report of the uplift rates calculated based on these shoreline angle altitudes (see Sect. 3.3.2). 4) Sedimentary logs of the terrace deposits at the OSL sampling sites. All maps are also provided as KMZ and shapefiles whith a metadata description (Normand et al., 2018). Additional comments on the terraces geomorphology and mapping choices are found as supplementary data text C.

## 4.2 Dating

### 4.2.1 Radiometric dating

Results of radiometric dating of the terraces can be found in Table 1. The radiocarbon ages measured on our samples span 20-50 ka, similar to previous radiocarbon dates in the Makran (see supplementary data Table A). There does not seem to be any kind of relationship between the altitude and radiocarbon age of the dated terraces (Fig. 8). However, radiocarbon results are

close to the limit of the method (>20 ka) and therefore caution should be taken in the interpretation of such ages.

$^{230}$Th/U dating results have to be considered as minimum ages because the corresponding $^{234}$U/$^{238}$U ratio is greater than that of seawater, indicating post-sedimentary Uranium incorporation from groundwater (Sect. 3.2.3) (Table 1). In the case of RN15-84 and RN15-90, $^{230}$Th/U ages are greater than $^{14}$C ages, therefore $^{14}$C ages do not represent true ages for these two samples. Sample RN15-87 is dated >16±0.14 ka with $^{230}$Th/U (minimum age) and 33.4±0.415 ka with $^{14}$C; in this case, the $^{14}$C age is

more relevant.

XRD on the aragonite samples showed that RN15-84 is undoubtedly recrystallized (~45% calcite) and therefore we rejected it. The two others contain more than 97% of Aragonite and in conjuncture with SEM images observations are considered reliable (Normand et al., 2018).

### 4.2.2 OSL dating

The results of OSL dating are presented in Table 2. The low values of overdispersion show that the aliquot measurements for each sample are consistent with each other. On the other hand, the effect of fading is quite significant for samples older than 100ky, with a correction of the age of up to 20%.

All results of OSL dating fall within, or close to a period of sea-level highstand (Fig. 9). Two samples are correlated to MIS 3, five samples fall within MIS 5a, three samples are correlated to MIS 5e, although slightly older, and finally two samples are

from MIS 7. At Lipar, two different levels at different altitude are both correlated to MIS 5a (Lipar T1 and T2). The main Chabahar-Ramin T1 terrace, sampled at different locations, yielded a MIS 5a age at the coastline and MIS 5e in the middle of

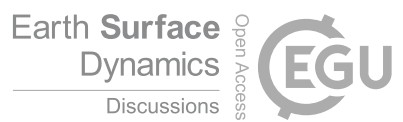

the terrace (Fig. 6a), possibly implying re-occupation of the terrace during the different substages of the last interglacial. The two northern Ramin samples (RN17-39 and RN17-40) are separated by an escarpment, but both fall within MIS 7. The Jask T1 terrace was dated at MIS 5a, which differs from the previous MIS 5e assignment of Page et al. (1979). The youngest terrace (Pasabander T1, samples RN17-24 and RN15-87) is dated 34.18±2.32 ka with OSL, which is coeval to its [14]C age of 33.4±0.42 ka.

## 4.3 Uplift rates

We focus on data from the eastern part of the Iranian Makran (from Tang to Pasabander) (Fig. 10), which is where most of the studied Iranian marine terraces are located. The terrace record is not spatially continuous along the coast, and we will discuss the possible missing links between the different terrace regions in Sect. 5.3. From the record of the studied terraces, we observe that the uplift rates vary spatially in a sinusoidal way between 0.2 and 1.5 mm y$^{-1}$, with an exceptionally high uplift rate in the easternmost portion of Iran (Pasabander), were MIS 3 terraces are emerged (Fig. 10). Terraces are tilted both eastwards and westwards with an angle of up to 0.73°. This trend seems to continue to Pakistan where the tilt of the Jiwani terraces mirrors that of those in Pasabander. A detailed summary of calculated uplift rates and their errors, based on shoreline angle altitudes extracted from the DEM and the eustatic curve of Spratt and Lisiecki (2016) are presented below each terrace map (Fig. 4c, 4h, 5c, 5f, 5i, 6c, 7f). Uplift curves overlapping each other indicates that a constant uplift scenario would fit through the terrace profile.

On the Makran coast, multiple normal faults offset terrace deposits (Fig. 2d, 2g) and modify uplift rates significantly. Heavily faulted regions include; the Ramin terraces, which seems to be built on the hanging wall of a ~15 km long south dipping normal fault system (Fig. 6a, 6b), and the Lipar terraces, where south-west dipping normal faults offset several terraces (Fig. 4f, 4g). Although some isolated normal faults also offset the terraces of Pasabander T3 and Gurdim, the remaining studied regions do not seem to be influenced by normal faulting. Minimum fault slip rates (based on the age of the terrace they offset) are reported in the uplift profiles (Fig. 4h, 5f, 6c, 7f) but remain relatively low (0.06-0.25 mm y$^{-1}$).

## 5. Discussion

### 5.1 Geochronological issues

Comparison between the results from the different dating methods are presented in Fig. 9. Within the same terrace, [230]Th/U minimum ages are in good correlation with the OSL dating, the latter being systematically older. However, [14]C ages are mostly considerably younger than results obtained with OSL or [230]Th/U, with one exception (Pasabander T1), discussed below. Moreover, we find no suitable morphostratigraphic interpretation based on radiocarbon ages, since there is no age-height relationship between the terraces of the sequence (Fig. 8). Therefore, we consider our [14]C ages to be minimum ages. We emphasize the need for caution when interpreting shell [14]C ages older than 20 ka, even if the shell is not seemingly recrystallized (e.g.: sample RN15-90) (e.g.: Bezerra et al., 2000; Busschers et al., 2014). We suggest that future marine terraces



studies should avoid the radiocarbon method on shells for terraces suspected to be older than the Holocene, or at least to corroborate radiocarbon ages greater than 20 ka with other methods, more suitable for longer timescales.

In the case of Pasabander T1, both [14]C and OSL agree towards a MIS 3a age, which is also compatible with the minimum [230]Th/U age of 16 ka (Fig. 9). Moreover, it is corroborated by the age of the overriding terrace (Pasabander T2, Sample RN17-

19), which was dated to a slightly older age of MIS 3c (53.54 ± 2.3 ka, see Table 2). Therefore, we consider an MIS 3 age for these two terraces reliable. MIS 3 terraces are scarce and controversial (Murray-Wallace and Woodroffe, 2014; Pedoja et al., 2014) because: (1) During this period, the sea-level was much lower than today (-50 to -110 m), implying that high uplift rates are needed to find them emerged today. (2) Due to the scarcity of MIS 3 terraces, our knowledge of sea-level behavior during this period is still lacking. For instance, the effect of Heinrich and Dansgaard–Oeschger events on sea-level variation is still

uncertain (Chappell, 2002; Hemming, 2004; Murray-Wallace and Woodroffe, 2014; Siddall et al., 2008). (3) Many [14]C dated terraces can be wrongly assigned to this period because it coincides with the limit of this method (Murray-Wallace and Woodroffe, 2014), as seen from the [14]C dating results reported in this paper and from previous work in the Makran (Fig. 9, supplementary data Table A).

## 5.2 Geological implications

The presence of MIS 3 terraces at Pasabander raises several geological issues. First, the region of Pasabander has an anomalously high uplift rate compared to the rest of Makran and compared to uplift rates along other subduction zones in the World (Henry et al., 2014; Pedoja et al., 2011). We will discuss the possible reasons for this anomaly in Sect. 5.3.2. Second, a high uplift rate does not fit with the terraces' altitude profiles. Considering that all antecedent terraces in a profile are also uplifting at the same time as the youngest terrace, it is not possible to fit our altitude profile with a straightforward uplift

history, unless we imply a subsidence event between MIS 3c and 3a (Fig. 11a), which seems unlikely. Similarly, by extrapolating an uplift rate of ~3.7 mm y$^{-1}$ calculated from the shoreline angle of T1, we should expect T2 (MIS 3c) and T3 (MIS 5a) shoreline angles to be at heights of 130 m and 270 m respectively (north of Beris village) (Fig. 11a), which is not observed. Third, along most of the Makran, the sea-floor topography offshore the terraces is currently nearly flat (<0.2° slope), which was plausibly also the case during past highstands. We believe that with such sea-floor topography, a lowstand (e.g.:

MIS 3) terrace should not be situated directly at the foot of a highstand terrace (e.g.: MIS 5a), but rather shifted laterally by several kilometers (Fig. 11b).

Therefore, although the data strongly support the existence of emerged MIS 3 terraces in Pasabander, the results are problematic when considered with other aspects. Our present hypothesis for T3 (an assignment to MIS 5a, the next logical suitable highstand) might be part of the issue and needs confirmation by further studies.

At Lipar, two terraces at different altitudes (T1 and T2) have both been dated to MIS 5a (samples RN17-27 and RN17-31 respectively) (Fig. 4f, 4j). In this region, several terraces are offset by normal faults, so it could be that both T1 and T2 are the same terrace offset by a normal fault. However, we have not observed any indication for the presence of such a fault in the field. Moreover, it is difficult to fit a fault in this geographical context without affecting the other terraces, which fit a constant



uplift scenario (Fig. 4h). Although both measures do not present any analytical abnormality, we believe the age of Lipar T2 sample (RN17-31) is questionable because the Lipar T2 uplift rate does not match the surrounding dated terraces (Fig. 4h). If assigned to MIS 5c (as can be expected from a terrace situated between two terraces dated MIS 5a and MIS 5e), its uplift rate would be compatible with other rates in the region (~0.5 mm y⁻¹).

Our results from the Ramin samples imply that the wide Ramin T1 has been built and reoccupied during the three substages of MIS 5 (Fig. 11c). We believe that the episodes of reoccupation during MIS 5e, 5c and 5a should be expressed by the succession of associated deposits, separated by flooding surfaces, as proposed by Jara-Muñoz and Melnick (2015). Unfortunately, we only have access to natural cross sections of the terrace deposits at the terrace borders, and we lack information from the center of Ramin T1 (also due to the presence of a large eolian field). Since the relative sea-level was

lower during MIS 5c and 5a than during MIS 5e, probably only the lower, seawards portion of the terrace was reoccupied (Fig. 11c). That would explain why the samples get younger towards the sea.

Another interesting point for Ramin T1 is its concave topographic profile (pink line, Fig. 6b, 6c). We believe this is due to the presence of a fault offshore from the Komb village (Fig. 6a). We mapped it as a potential southward dipping normal fault, but we did not observed it on the field. Note that a northward dipping reverse fault would produce a similar deformation pattern

and is also relevant in the MSZ convergence context. We believe that this deformation is only relevant locally, and it should not be considered when analyzing the general trends of lateral uplift variations along the Makran (Sect. 5.3.1, Fig. 10).

## 5.3 Tectonic implications

### 5.3.1 Terrace uplift variation

Along-strike variations in uplift rates are evidenced by east-west terrace tilting. We infer that this process is ongoing since the

development of the oldest terraces of each sequences. In those regions of the Iranian Makran where several terraces of the same profile are tilted (Konarak, Chabahar, Lipar, Pasabander), the shoreline angles altitude profiles get more and more steeply tilted as the terraces get older while the uplift profiles stay roughly parallel (constant uplift through time) (Fig. 4g-h, Fig. 5h-i, Fig. 6b-c, Fig. 7e-f). This is similar to what can be predicted to happen if tilting forces have been continuous through time (Fig. 12a) (Lajoie, 1986). Moreover, we observe that the uplift rates vary from west to east with a wavelength of roughly 20-

30 km (Fig. 10). We can therefore conclude that although the forces responsible for terrace uplift vary spatially, they are roughly steady through the Late Pleistocene (evidenced by constant uplift rates) and have been continuously active throughout the buildup of the Makran sequences of marine terraces (evidenced by terrace tilting increasing with age).

Internal structures of the outcropping sedimentary prism of Makran can easily be seen with satellite images. Notably, in the Makran, we observe E-W striking, doubly plunging anticlines and synclines. Although the east-west wavelength of such

structures are large in the Pakistani Makran (50-80+ km), plunging anticlines and synclines in the Iranian Makran have a shorter wavelength, comparable to that of the tilted terraces (Burg et al., 2012; Farhoudi and Karig, 1977; Leggett and Platt, 1984). Hence, terrace tilting might be related to the active growth of folds within the sedimentary prism (e.g.: Melnick et al.,





2006) as already observed in a north-south direction by a study of fluvial terraces in the Iranian Makran (Haghipour et al., 2012).

Normal faults have shown to be an important actor of terrace deformation in the Makran. Their influence on terrace uplift rates has already been observed in different subduction zones around the world (Gallen et al., 2014; Meschis et al., 2018; Roberts

et al., 2013; Saillard et al., 2011). The observation of normal faulting in a convergence context is intriguing. On the one hand, marine terraces indicate that the region is undergoing long-term uplift, while the normal faults alone might suggest subsidence. Further work is needed to understand how the normal faults function in a region that experiences long-term compression. At this stage, we suspect that the normal faults might be related to post-seismic relaxation following large megathrust earthquakes, as has also been observed in other subduction zones of the world (Hardebeck, 2012; Hasegawa et al., 2011). Since normal

faulting is common in the coastal Makran, it is possible that uplift rate variation along the coast is due to occurrence of major normal faults between the headlands hosting the terraces. Although their physical expression in the landscape is not visible, such structures have already been observed close to the coast in offshore seismic sections (Grando and McClay, 2007; Harms et al., 1984; Mokhtari et al., 2008).

### 5.3.2 Anomalously high uplift rates and seismic segmentation

We have expressed our concerns regarding the ambiguous geological implications of MIS 3 terraces at Pasabander (Sect. 5.2). However, it is necessary to explore the possible significance of such high uplift rates if these results are correct.

In the Andean subduction zone, where seismic and geomorphological data are abundant, regions of higher uplift rates have been noted to coincide with the limits of earthquake rupture areas (Jara-Muñoz et al., 2015; Saillard et al., 2017). In the Makran, it has often been proposed that the subduction interface might be seismically segmented between east and west, the limit being

the offshore Sonne fault (Kukowski et al., 2000). The Pasabander region, showing anomalously high uplift rates, could be the surficial expression of a portion of the subduction interface experiencing strong interseismic coupling, effectively acting as seismic barrier (e.g.: Saillard et al., 2017). On the other hand, other factors, such as underplating of sediments (Grando and McClay, 2007; Kopp et al., 2000; Platt et al., 1985), subduction of topographic anomalies (Henry et al., 2014; Spikings and Simpson, 2014), shale diapirism (Grando and McClay, 2007) or the action of local thrusts have also been shown to produce

similar surficial response. Future research should focus on confirming the age of these terraces, understanding the processes responsible for the anomalously high uplift rate recorded in the Pasabander region as well as complementing the uplift map of the Makran with data from the more seismically active eastern segment of the MSZ (Pakistan).

### 5.3.3 The role of subduction dynamics

Deformation recorded on the overriding plate of a subduction zone is usually associated with subduction dynamics. In this

regard, marine terraces provide information on the cumulative coseismic and interseismic signals accumulated over a time scale of several seismic cycles (Burbank and Anderson, 2001; Keller and Pinter, 2002; Lajoie, 1986; Melnick et al., 2006; Sato and Matsu'ura, 1992; Simpson, 2015). The high uplift rates recorded in the Iranian Makran indicate that the upper plate has





been accumulating considerable permanent deformation during the Late Quaternary. At this stage, it is not clear exactly how that deformation is being accommodated, and whether, for example, it is associated with large earthquakes, but it remains a possible explanation (Rajendran et al., 2013; Schellart and Rawlinson, 2013; Smith et al., 2013).

It has been pointed out by previous work that deformation in the upper plate can significantly affect marine terraces' uplift rates and lead to inadequate conclusions regarding subduction dynamics (Meschis et al., 2018). Most of the present MSZ convergence is apparently accommodated at the tip of the prism, where imbricate thrusts are currently developing (Ellouz-Zimmermann et al., 2007; Farhoudi and Karig, 1977; Grando and McClay, 2007; White and Louden, 1982; White and Ross, 1979). Nevertheless, our results indicate that the internal part of the accretionary prism is still accommodating significant internal deformation and uplift, as also shown by Haghipour et al. (2012). Those few terraces that are not tilted or faulted (e.g.: Jask, Tang, Gurdim, Astola, Ras Shamal Bandar) might provide the best insights on the uplift component directly linked to subduction dynamics and earthquake cycles.

## 6. Conclusion

We have studied the uplifted marine terraces of the Iranian Makran using a combination of field mapping, remote sensing and dating with the $^{14}$C, $^{230}$Th/U and OSL methods. The dating results indicate that radiocarbon dating on shells greater than 20 ka does not always coincide with the other methods, and therefore we recommend future marine terrace studies to supplement similar results with other dating techniques. The studied terraces ages suggest that they formed between marine isotope stages 3 and 7. Undated terraces might date back to MIS 17 at Chabahar, where the most complete terrace sequence is situated. We also identified the global geodynamic marker, the MIS 5e benchmark, along the 200 km long coastal strip between Pasabander and Tang. Uplift rates of the terraces vary significantly along strike to the trench between 0.05 and 1.2 mm y$^{-1}$. Emerged terraces at the easternmost boundary of Iran have been dated to MIS 3, implying an exceptionally fast rate of up to 3-5 mm y$^{-1}$. Based on geological observations and morphological relations between terraces from the sequence, we have difficulties accepting these results as is. If true, such local rates are amongst the highest worldwide and are possibly linked to strong interseismic coupling at the subduction interface. Although the upper plate is currently accommodating considerable internal deformation, we do not know yet if this uplift occurs in a continuous manner or episodically, during large megathrust earthquakes.

## 8 Data availability

Supplementary data for this paper are found in the following repository: 10.5281/zenodo.1468857 (Normand et al, 2018). They come in four forms: A) Terrace maps GIS, including mapping information and confidence indexes (data model description is also included). Both KMZ and SHP files are available for Google Earth and other GIS software. B) Supplements for the Radiocarbon and $^{230}$Th/U dating method (such as analytical details, XRD results and SEM images of shell samples). C)

Supplements for the OSL dating method (such as analytical details, Environmental dose calculation, fading tests results). D) Additional field pictures supplementing Fig.2.

## 13 Author contribution

Raphaël Normand did the fieldwork, lab work, wrote the paper and created the figures. Guy Simpson accompanied him in the field and had substantial input on the tectonic and geodynamic aspects. Frédéric Herman and Rabiul H. Biswas contributed in OSL dating; sample preparation, lab work, and results processing. Abbas Bahroudi provided his input on Iranian geology and tectonics and organized field campaigns. Bastian Schneider provided the tan-DEM of the studied region and contributed to the discussion on terrace morphologies.

## 14 Competing interests

The authors declare that they have no conflict of interest.

## 17 Acknowledgements

This work was funded by the Swiss National Science Foundation, project n°200021_155904. We thank Kevin Pedoja for his input and all the fruitful and interesting discussions. We are also grateful to Reza Ensani, Feisal Arjomandi, Nurrudin Mazarzehi, Yousef Adeeb and Gholamreza Hosseinyar for helping us with logistics in Iran and accompanying us in the field. We also would like to thank the analytical assistance of Bassam Ghaleb for [230]Th/U dating, Agathe Martignier for SEM, Annette Süssenberger and Emanuelle Ricchi for XRD analysis.

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



**Table 1. Results of radiometric dating**

| Sample | Terrace | Alti. [m] | Mineralogy | Radiocarbon age* Conventional ± 1σ BP | Radiocarbon age* Calibrated ± 2σ Cal BP | XRD Arag.-Calcite norm. % | 230Th/U age 230Th/U age ± 2σ [ka] | 230Th/U age 234U/238U init ± 2σ | Comments |
|---|---|---|---|---|---|---|---|---|---|
| RN15-84 | STRUCTURAL | 206 | Aragonite | 28650 ± 140 | 31856 ± 495 | 54.58 - 45.42 | 173.25 ± 3.64 | 1.51 ± 0.01 | Recrystallized |
| RN15-87 | PASABANDER T1 | 49 | Aragonite | 29860 ± 170 | 33400 ± 415 | 99.7 - 0.3 | 16 ± 0.14 | 1.32 ± 0.01 | [14]C age reliable |
| RN15-90 | LIPAR T1 | 12 | Aragonite | 43280 ± 170 | 46137 ± 1417 | 96.97 - 3.03 | 57.27 ± 0.69 | 1.27 ± 0.01 | Th/U min. age > [14]C age |
| RN15-53 | RAMINT1 | 59 | Calcite | 28200 ± 110 | 31337 ± 208 | | | | Minimum age |
| RN15-92 | LIPAR T3 | 66 | Calcite | 26060 ± 110 | 29495 ± 372 | | | | Minimum age |
| RN15-91 | RAMIN T1 | 49 | Calcite | 43080 ± 360 | 45734 ± 702 | | | | Minimum age |
| RN15-54 | CHABAHAR T? | 64 | Calcite | 38790 ± 240 | 42331 ± 366 | | | | Minimum age |
| RN15-85 | CHABAHAR T3 | 117 | Calcite | 38670 ± 240 | 42259 ± 368 | | | | Minimum age |
| RN15-86 | CHABAHAR T3 | 110 | Calcite | 40780 ± 290 | 43770 ± 604 | | | | Minimum age |
| RN15-20 | CHABAHAR T5 | 81 | Calcite | > 43500 | | | | | Minimum age |
| RN15-57 | CHABAHAR T5 | 95 | Calcite | 41700 ± 310 | 44580 ± 640 | | | | Minimum age |
| RN15-59 | CHABAHAR T6 | 108 | Calcite | > 43500 | | | | | Minimum age |
| RN15-64 | CHABAHAR T6 | 140 | Calcite | 38290 ± 230 | 42013 ± 370 | | | | Minimum age |

\* Calibrated using Oxcal 4.2 (Bronk Ramsey and Lee, 2013), with the curves IntCal 13 and Marine 13 (Reimer et al, 2013)

5 Reservoir correction, Delta_R = 236±31 years for Makran, according to the website, http://calib.org/marine/

All Calcitic shells have ages close to the 14C limit.

It is not possible to check the recrystallisation stage of the calcitic shells using XRD, therefore all these ages have to be considered minmum age.



**Table 2. Results of OSL dating**

| Sample | Terrace | Alti. | Sample depth | Paleodose CAM ± 1σ | N°of Aliquots | RSD | OD |
|---|---|---|---|---|---|---|---|
| | | [m] | [m] | [Gy] | out of 12 | % | % |
| RN17-24 | PASABANDER T1 | 33 ± 2 | 1 ± 0.2 | 65.76 ± 0.61 | 12 | 3.33 | 1.6 |
| RN17-19 | PASABANDER T2 | 24 ± 2 | 1 ± 0.2 | 111.36 ± 3.13 | 12 | 10.30 | 9.2 |
| RN17-27 | LIPAR T1 | 20.5 ± 2 | 0.5 ± 0.1 | 104.8 ± 1.48 | 12 | 5.13 | 3.9 |
| RN17-31 | LIPAR T2 | 50.5 ± 2 | 0.5 ± 0.1 | 131.65 ± 3.57 | 12 | 10.38 | 8.8 |
| RN17-30 | LIPAR T3 | 58.5 ± 2 | 1.5 ± 0.2 | 219.6 ± 3.48 | 12 | 5.71 | 3.9 |
| RN17-14 | RAMIN T1 | 3 ±2 | 1 ± 0.3 | 141.59 ± 5.06 | 12 | 13.08 | 11.9 |
| RN17-43 | RAMIN T1 | 59.5 ± 2 | 0.5 ± 0.1 | 373.1 ± 19.9 | 11 | 21.65 | 17 |
| RN17-39 | RAMIN T2 | 41 ± 2 | 1 ± 0.2 | 343.81 ± 9.57 | 12 | 9.95 | 8.7 |
| RN17-40 | RAMIN T2 | 72.5 ± 2 | 0.5 ± 0.1 | 296.96 ± 7.44 | 12 | 9.01 | 7.8 |
| RN17-45 | GURDIM T1 | 64 ± 2 | 0.5 ± 0.1 | 143.51 ± 2.29 | 12 | 5.63 | 4.5 |
| RN17-47 | JASK T1 | 2.5 ± 2 | 0.5 ± 0.1 | 120.32 ± 2.17 | 12 | 6.50 | 5.3 |
| RN17-48 | KONARAK T3 | 32.3 ± 2 | 1.7 ± 0.2 | 244.07 ± 9.23 | 12 | 14.52 | 12.4 |





| U | Th | K | Rb | Water content | Environmental dose | Uncorrected Age ± 1σ | g-value | Fading corrected Age ± 1σ | Assigned MIS |
|---|---|---|---|---|---|---|---|---|---|
| ppm | ppm | % | ppm | % | [Gy ka$^{-1}$] | [ka] | % per decade | [ka] | |
| 2.4 ± 0.1 | 2.5 ± 0.09 | 0.38 ± 0.011 | 15 ± 1.15 | 2 ± 2 | 1.92 ± 0.13 | 34.18 ± 2.3 | 0 ± 0 | 34.18 ± 2.3 | 3a |
| 2.5 ± 0.1 | 3.4 ± 0.13 | 0.53 ± 0.015 | 20 ± 1.54 | 2 ± 2 | 2.18 ± 0.13 | 51.01 ± 3.4 | 0.72 ± 0.40 | 53.54 ± 3.9 | 3c |
| 1.8 ± 0.08 | 0.6 ± 0.02 | 0.15 ± 0.004 | 3 ± 0.23 | 2 ± 2 | 1.37 ± 0.12 | 76.77 ± 6.9 | 0.69 ± 0.41 | 80.6 ± 7.55 | 5a |
| 3 ± 0.13 | 1.4 ± 0.05 | 0.22 ± 0.006 | 8 ± 0.62 | 2 ± 2 | 1.86 ± 0.14 | 70.62 ± 5.5 | 0.60 ± 0.42 | 73.67 ± 6.15 | 5a |
| 2.7 ± 0.11 | 1.5 ± 0.06 | 0.31 ± 0.009 | 10 ± 0.77 | 2 ± 2 | 1.84 ± 0.13 | 119.35 ± 8.7 | 1.19 ± 0.40 | 130.28 ± 10.42 | 5e |
| 2.3 ± 0.1 | 2 ± 0.07 | 0.37 ± 0.011 | 14 ± 1.08 | 2 ± 2 | 1.83 ± 0.13 | 77.27 ± 6.1 | 0 ± 0 | 77.27 ± 6.1 | 5a |
| 5.9 ± 0.25 | 2.3 ± 0.09 | 0.46 ± 0.014 | 17 ± 1.31 | 2 ± 2 | 3.06 ± 0.18 | 121.90 ± 9.8 | 1.86 ± 0.46 | 140.21 ± 12.47 | 5e |
| 3.2 ± 0.13 | 1.8 ± 0.07 | 0.38 ± 0.011 | 12 ± 0.92 | 2 ± 2 | 2.1 ± 0.14 | 163.61 ± 11.7 | 2.15 ± 0.41 | 193.25 ± 15.58 | 7 |
| 2.6 ± 0.11 | 1.2 ± 0.04 | 0.20 ± 0.006 | 6 ± 0.46 | 2 ± 2 | 1.71 ± 0.13 | 173.64 ± 13.9 | 2.84 ± 0.43 | 218.36 ± 19.54 | 7 |
| 1.6 ± 0.08 | 1.9 ± 0.07 | 0.52 ± 0.015 | 16 ± 1.23 | 2 ± 2 | 1.8 ± 0.12 | 79.90 ± 5.6 | 1.50 ± 0.42 | 89.00 ± 6.89 | 5a |
| 1.5 ± 0.06 | 1.3 ± 0.05 | 0.47 ± 0.014 | 8 ± 0.62 | 2 ± 2 | 1.66 ± 0.12 | 72.45 ± 5.4 | 2.08 ± 0.41 | 84.45 ± 7.05 | 5a |
| 3.2 ± 0.13 | 1.6 ± 0.06 | 0.34 ± 0.01 | 12 ± 0.92 | 2 ± 2 | 2.03 ± 0.14 | 120.39 ± 9.4 | 2.10 ± 0.42 | 141.30 ± 12.09 | 5e |



**Table 3. Uplift rates calculation at the shoreline angle situated directly northwards of each sample. We present here the results calculated with two sea-level curves (Spratt and Lisiecki, 2016, Shakun et al., 2015), though any other curve could be used in the same manner, following the method described in Sect. 3.3. Notice that these uplift rates are only relevant locally for tilted terraces, thus, a record of spatial uplift rate variation is reported as a graph below each terrace map (Fig. 4-7).**

| Sample | Terrace | Type of Marker[a] | Sediment thickness[b] | Shoreline angle elevation | Assigned MIS (Table 2) | Assigned MIS age[c] | Apparent uplift rate $U_a = E/A$ | | |
|---|---|---|---|---|---|---|---|---|---|
| in eqs. (1-3) | | | | $E \pm \Delta E$ | | $A \pm \Delta A$ | Umax | Umin | $U_a \pm \Delta U_a$ |
| | | | [m] | [m] | | [ka] | [mm y⁻¹] | | [mm y⁻¹] |
| RN17-24 | PASABANDER T1 | SA | 2 ± 0.5 | 35 ± 2.1 | 3a | 34 ± 5 | 1.28 | 0.84 | 1.06 ± 0.22 |
| RN17-19 | PASABANDER T2 | ESA | 2.5 ± 0.5 | 24.5 ± 2.1 | 3c | 51 ± 5 | 0.58 | 0.40 | 0.49 ± 0.09 |
| RN17-27 | LIPAR T1 | SA | 2 ± 0.5 | 20 ± 2.1 | 5a | 80 ± 5 | 0.29 | 0.21 | 0.25 ± 0.04 |
| RN17-31 | LIPAR T2 | SA | 8 ± 0.5 | 45 ± 2.1 | 5a | 80 ± 5 | 0.63 | 0.51 | 0.57 ± 0.06 |
| RN17-30 | LIPAR T3 | ESA | 6 ± 0.5 | 67 ± 2.1 | 5e | 122 ± 6 | 0.60 | 0.51 | 0.55 ± 0.04 |
| RN17-14 | RAMIN T1 | BSA | 6 ± 4 | -2 ± 4.3 | 5a | 80 ± 5 | 0.03 | -0.07 | -0.02 ± 0.05 |
| RN17-43 | RAMIN T1 | ESA | 2.5 ± 0.5 | 57.5 ± 2.1 | 5e | 122 ± 6 | 0.51 | 0.43 | 0.47 ± 0.04 |
| RN17-39 | RAMIN T2 | BSA | 6 ± 4 | 41 ± 4.5 | 7a | 196 ± 6 | 0.24 | 0.18 | 0.21 ± 0.03 |
| RN17-40 | RAMIN T2 | ESA | 4 ± 0.5 | 73 ±2.1 | 7c | 211.5 ± 5.5 | 0.36 | 0.33 | 0.35 ± 0.02 |
| RN17-45 | GURDIM T1 | ESA | 3 ± 0.5 | 62 ± 2.1 | 5a | 80 ± 5 | 0.85 | 0.71 | 0.78 ± 0.07 |
| RN17-47 | JASK T1 | BSA | 6 ± 4 | -3 ± 4.3 | 5a | 80 ± 5 | 0.02 | -0.09 | -0.03 ± 0.05 |
| RN17-48 | KONARAK T3 | ESA | 2.5 ± 0.5 | 32.5 ± 2.1 | 5e | 122 ± 6 | 0.30 | 0.24 | 0.27 ± 0.03 |

[a]SA = Shoreline angle. ESA = Eroded shoreline angle. BSA = Buried shoreline angle; shoreline angle position and altitude unknown, uplift rates calculated from sample elevation

[b]Sediment thickness at the shoreline angle, as observed on the field. 6 ± 4 m is set when the thickness could not be determined

[c]See supplementary data B.2

[d]Using eq. 1: U = (E-e)/A  (Lajoie, 1986)





| Eustatic estimations Spratt and Lisiecki, 2016 | | | Uplift ranges. Spratt and Lisiecki, 2016 | | | | | Eustatic estimations Shakun et al., 2015 | | Uplift ranges. Shakun et al., 2015 | | | | |
|---|---|---|---|---|---|---|---|---|---|---|---|---|---|---|
| $e$ | $+\Delta e$ | $-\Delta e$ | Umax | Umin | $U^d$ | $+\Delta U$ | $-\Delta U$ | $e$ | $\pm\Delta e$ | Umax | Umin | $U^d$ | $+\Delta U$ | $-\Delta U$ |
| [m] | [m] | [m] | [mm y$^{-1}$] | | | [mm y$^{-1}$] | | [m] | [m] | [mm y$^{-1}$] | | | [mm y$^{-1}$] | |
| -97.5 | 10.5 | 14.5 | 3.08 | 5.14 | 3.90 | 1.24 | 0.82 | -78 | 12 | 2.54 | 4.38 | 3.32 | 1.06 | 0.79 |
| -63 | 8 | 19 | 1.38 | 2.36 | 1.72 | 0.64 | 0.33 | -55 | 12 | 1.17 | 2.03 | 1.56 | 0.48 | 0.39 |
| -25.5 | 11.5 | 22.5 | 0.38 | 0.93 | 0.57 | 0.37 | 0.19 | -22 | 15 | 0.29 | 0.79 | 0.53 | 0.26 | 0.23 |
| -25.5 | 11.5 | 22.5 | 0.67 | 1.27 | 0.88 | 0.39 | 0.21 | -22 | 15 | 0.59 | 1.12 | 0.84 | 0.28 | 0.25 |
| 3 | 12 | 20 | 0.39 | 0.74 | 0.52 | 0.22 | 0.13 | -10 | 10 | 0.51 | 0.77 | 0.63 | 0.14 | 0.12 |
| -25.5 | 11.5 | 22.5 | 0.09 | 0.67 | 0.29 | 0.38 | 0.20 | -22 | 15 | 0.01 | 0.52 | 0.25 | 0.27 | 0.24 |
| 3 | 12 | 20 | 0.32 | 0.66 | 0.45 | 0.21 | 0.13 | -10 | 10 | 0.43 | 0.69 | 0.55 | 0.13 | 0.12 |
| -55.5 | 13.5 | 21.5 | 0.19 | 0.40 | 0.25 | 0.15 | 0.06 | -68 | 16 | 0.18 | 0.41 | 0.29 | 0.12 | 0.11 |
| -55.5 | 13.5 | 21.5 | 0.27 | 0.53 | 0.38 | 0.15 | 0.11 | -68 | 16 | 0.39 | 0.55 | 0.47 | 0.08 | 0.08 |
| -25.5 | 11.5 | 22.5 | 0.87 | 1.49 | 1.09 | 0.40 | 0.22 | -22 | 15 | 0.79 | 1.35 | 1.05 | 0.30 | 0.26 |
| -25.5 | 11.5 | 22.5 | 0.08 | 0.66 | 0.28 | 0.38 | 0.20 | -22 | 15 | 0.00 | 0.51 | 0.24 | 0.27 | 0.24 |
| 3 | 12 | 20 | 0.12 | 0.44 | 0.24 | 0.20 | 0.12 | -10 | 10 | 0.24 | 0.47 | 0.35 | 0.12 | 0.11 |



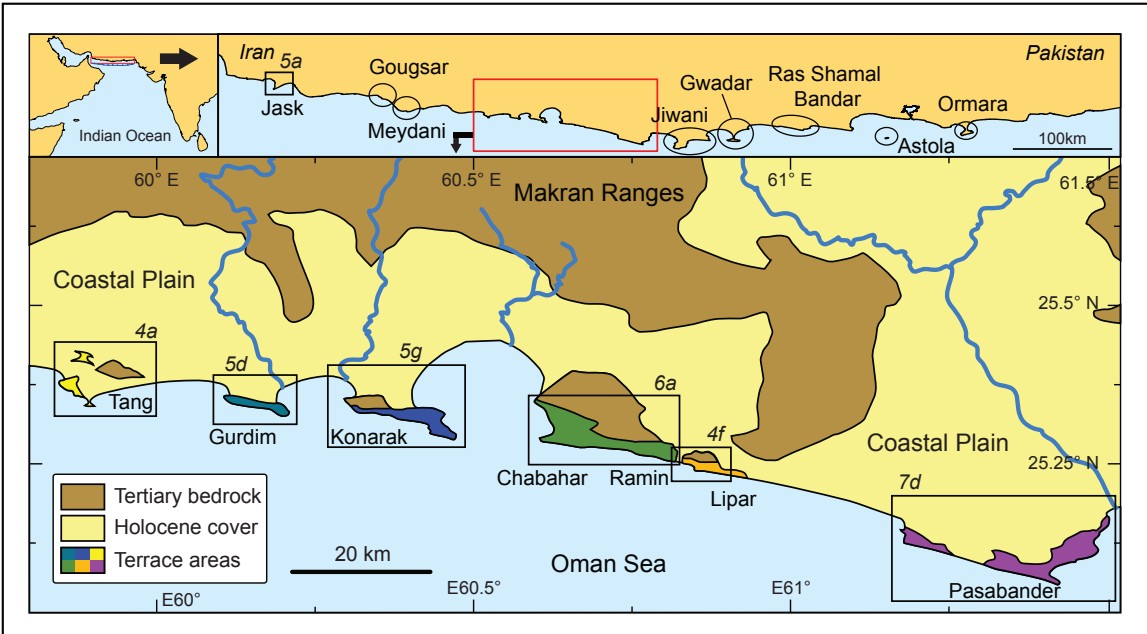

Figure 1. Map of the studied areas. Black squares: Terraces studied, visited and mapped in this paper (Fig. 4-7, italic numbers). Black circles: Other occurrences of marine terraces along the Makran Subduction zone.

Figure 2. Picture and sketches presenting the terrace geomorphology and sedimentology. Additional field pictures can be found in Normand et al. (2018). a) Sketch of the general morphology of Makran terraces and the different criteria of recognition. Method used to calculate E±ΔE: Alti±ΔAlti is the altitude obtained from the DEM or the GPS measurement on the field, S±ΔS is the sediment thickness obtained from field observations, or we used 6±4 m as a general value when the real sediment thickness was not directly observed. b) View of Lipar T1 (lower) and T3 (upper) (shot from 25.247924° N, 60.846619° E, looking southeast). We can see the waves carving the current terrace into both T1 and T3. Boulders of terrace deposits and bedrock origin are being sedimented within the beach deposits at the feet of the cliff. c) Close up of the terrace deposit of Lipar T2 (25.249823° N, 60.839074° E). The associated stratigraphic log is the number 1, reported in Fig. 4i. Notice the boulders at the base and the prograding sequence from shoreface to the foreshore parallel lamination at the top. d) Outcrop at Lipar lake, showing numerous south dipping normal faults cross cutting the tertiary bedrock, and sometimes even the topping horizontal layer of terrace deposit (Ramin T1) (25.260840° N, 60.830019° E). e) Modern boulders, i.e. boulders of bedrock and terrace deposit material being sedimented in modern beach deposits at the feet of the cliff (25.253043° N, 60.811024° E). d) Their paleo equivalent, at 150 m of altitude, now buried in cemented beach deposits (25.335013° N, 60.640952° E). Notice the sedimentary structures indicating that the boulder is tilted, similar to the modern boulders (rockfall). g) Surficial expression of a northward dipping normal fault, conjugate to those showed in Fig. 2d (25.258126° N, 60.824896° E). h) View of the back of Ramin T2 and its eroded shoreline angle (shot from 25.286121° N, 60.769900° E, looking east). Notice the angular unconformity with the tertiary bedrock, which composes most of the foreground. The main surface visible on the right of the picture is Ramin T1. Lipar headland is visible in the background. i) and j) Road cut cross section through a paleocliff (25.332877° N, 60.621057° E). We can see the unconformity with the bedrock bedding (paleocliff - green line) and the rockfall boulders sedimented within the terrace deposits.









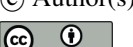

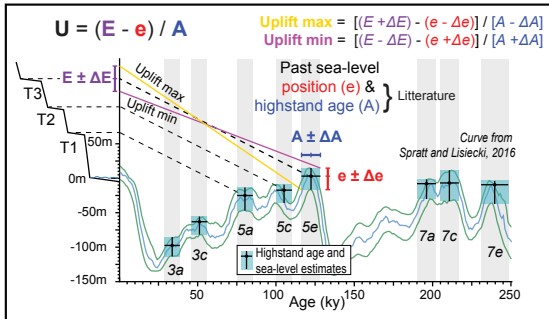

**Figure 3. Uplift calculation method. The curve shown is PCA 1 with bootstrapping 2.5 and 97.5 percentile from Spratt and Lisiecki (2016). Light grey bands are the MIS ages from the literature reported in supplementary data Table B.2.**

**Figure 4. Terrace maps of the Tang and Lipar regions (see Fig. 1). a) Terrace map of Tang. b) Shoreline angle altitude in an east-west profile. c) Uplift rates. d) Altitude profile through the terraces (profile A-B). e) Legend for all terrace maps, sedimentary logs and altitude / uplift profiles. f) Terrace map. g) Shoreline angle altitude in an east-west profile. h) Uplift rates. i) Sedimentary log at sampling locations and sample OSL age (blue). j) Altitude profile through the terraces (profile C-D).**





**Figure 5.** Terrace maps of the Jask, Gurdim and Konarak regions (see Fig. 1). a) Terrace map of Jask. b) Shoreline angle altitude in an east-west profile. c) Uplift rates. d) Terrace map of Gurdim. e) Shoreline angle altitude in an east-west profile. f) Uplift rates. g) Terrace map of Konarak. h) Shoreline angle altitude in an east-west profile. i) Uplift rates. j) Sedimentary log at sampling locations and sample OSL age (blue). k) Altitude profiles through the terraces.

**Figure 6.** Terrace map of the Chabahar and Ramin regions (see Fig. 1). a) Terrace map. b) Shoreline angle altitude in an east-west profile. c) Uplift rates. d) Altitude profile through the terraces. e) Sedimentary log at sampling locations and sample OSL age (blue).





Figure 7. Terrace map of the Pasabander region (see Fig. 1). a) Altitude profile through the terraces. b) Sedimentary log at sampling locations and sample OSL age (blue). c) Terrace map of the Kalani terrace, situated a few kilometers northwards of Gavatre. d) Terrace map of Pasabander. e) Shoreline angle altitude in an east-west profile. f) Uplift rates.





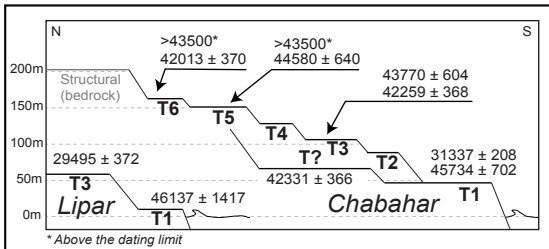

**Figure 8.** [14]C dating on Chabahar-Ramin and Lipar terraces. There is no morphostratigraphic relationship between the terraces ages and altitude if we consider the [14]C ages to be finite.





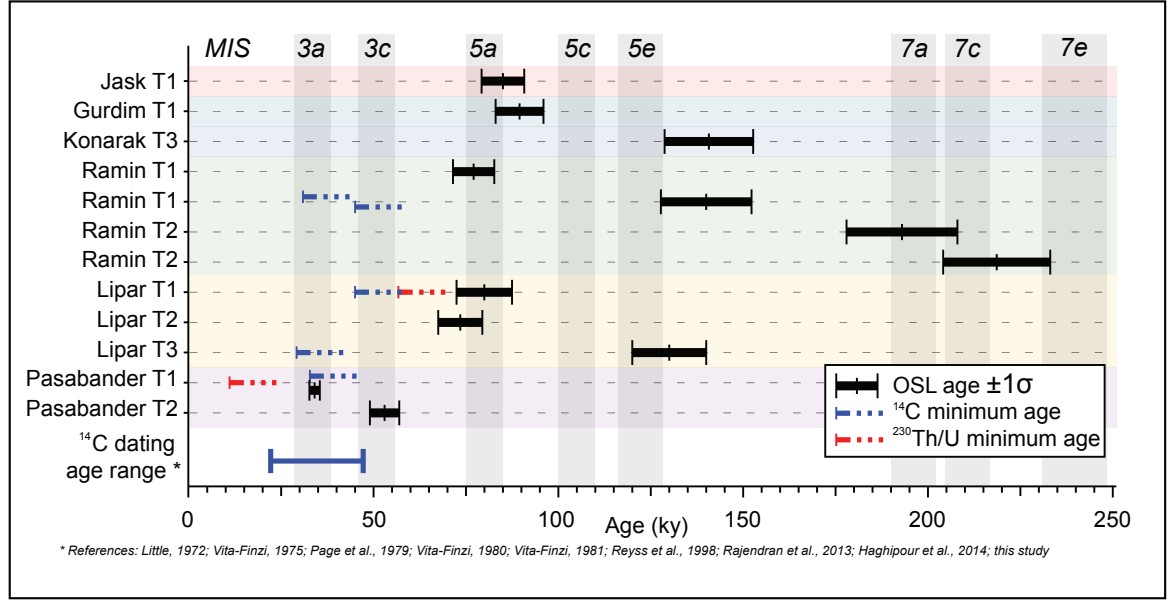

**Figure 9. Comparison of age results for the different Makran marine terraces. Grey vertical bands: Age extent of the different sea-level highstands (supplementary data Table B.2). For the list of previous ¹⁴C dating in the Iranian Makran, see supplementary data Table A.**



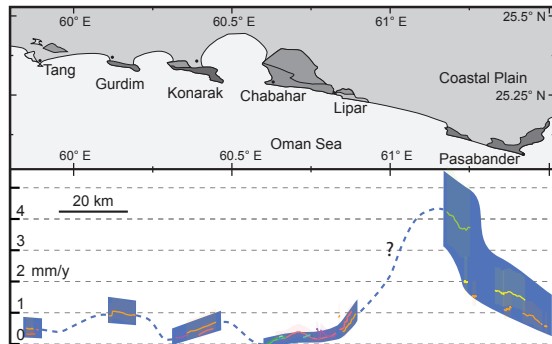

**Figure 10. Uplift rate spatial variation along the studied region. Colored lines are uplift rates reported from the different maps (Fig. 4-7). Dark blue polygons are uplift rates and associated errors in regions where marine terraces outcrop. Blue dashed lines are inferred uplift rate behavior between terrace areas. Between Lipar and Pasabander, the change in uplift rates is strong and thus subject to variations.**



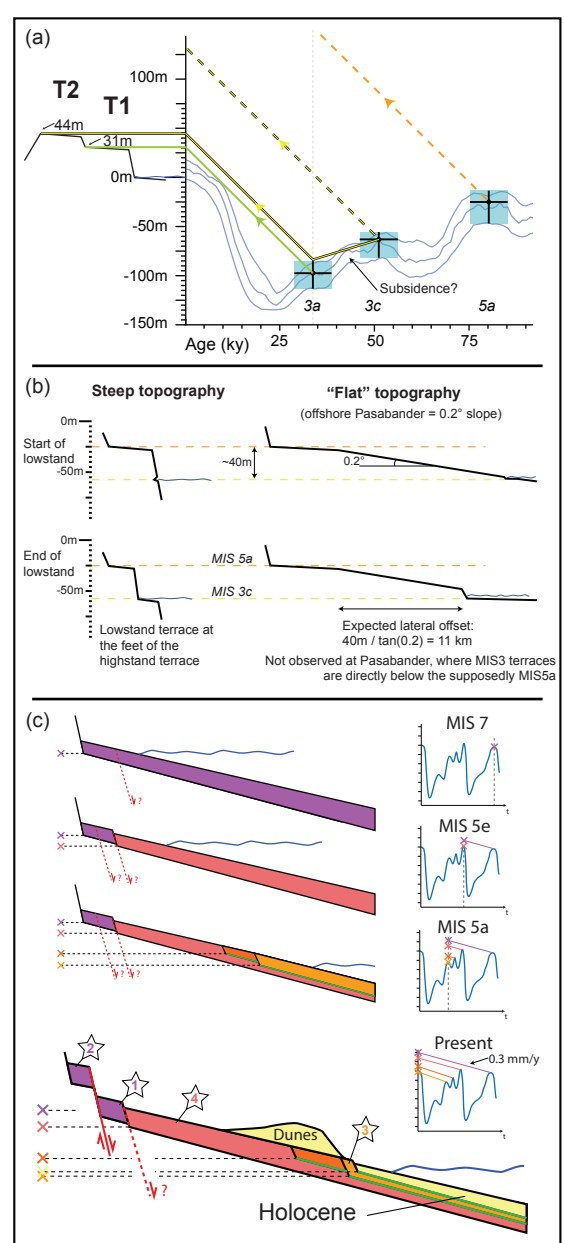

**Figure 11. Geological implications and issues associated with the dating results. a) Uplift rate history of Pasabander T1 and T2 at Beris village (profile A-B in Fig. 7a) (see text). Sea level curve from Spratt and Lisiecki (2016). b) Profiles perpendicular to the shoreline showing the expected development of a lowstand terrace with steep and not steep (as is offshore Pasabander) sea-floor topography. Uplift effect on relative sea-level is ignored for simplicity. c) Sketch of a possible terrace carving / re-occupation and sediment deposition scenario for the main Chabahar-Ramin T1 (Profile C-D, Fig. 6d), with an uplift rate of ~0.3 mm y⁻¹. Inset = sea-level history. Throughout the successive highstands, the sea re-occupies only the lower portion of the terrace as the land uplifts. The position of the samples and sedimentary logs presented in Fig. 6e are added. Flooding surfaces within the sedimentary deposits are represented as green lines. Vertical scale exaggerated.**




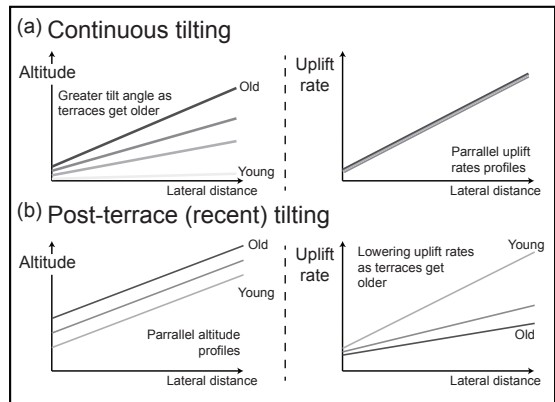

**Figure 12. Terrace tilting scenarios.** Expected shoreline angle altitude and uplift profiles in the scenarios of a) continuous tilting before and after terrace emplacement (as in Makran) and b) recent tilting, after the youngest terrace emplacement (not observed in Makran).