# Peer review of "Dating and morphostratigraphy of uplifted marine terraces in the Makran subduction zone (Iran)"

_Earth Surface Dynamics, 2018_

## Referee Comment (RC1) · Regard (Referee) · 5 Dec 2018

Article contents This work focusses on marine terraces as uplift markers along the Makran coast, in SE Iran/SW Pakistan. The Makran is an actively deforming emerged accretionary prism, developing over the north-dipping subduction of the Arabian plate. The work is based on ∼13 study sites distributed over hundreds of kilometers along the E-W coast. The data is made of geomorphic, tectonic and stratigraphic observations of marine terrace succession and organization. Dating attempts are presented with various methods (OSL, C-14 and U/Th).

General comments This is a valuable detailed work on marine terraces along the

[Figure]

Makran's coast. Unfortunately, the dating constraints are not always clear, but the authors made an interesting work in explaining quite clearly how these apparently confusing dates may be coherent, as evidenced by constant uplift rates through time. A striking point is the presence of unusual MIS3 terraces, supported by two scattered OSL ages (34 and 51kaBP). The data is not enough to convince me, but I have no alternative hypothesis. Thus I advise the authors to explore the possibility OSL dates are erroneous, in order to strengthen their conclusions. My main criticism comes from tectonics. The Makran is an active accretionary prism, probably hosting important thrusts. The authors only write about normal faults with limited Pleistocene offsets. The current tectonical framework is too light to understand the relationships between faults (either normal/inverse) and the uplift rate distribution. I think this problem could be addressed through mapping the main structures in figure 1 (both parts) and improving the background and discussion. I have some other comments about the timing (see thereafter). Apart to these criticisms I have been impressed by the quality of illustrations. There are lot of information reported: this paper must become a reference for future studies in the area. I support the publication of this manuscript.

Major comments. About tectonics. My experience at the western edge of the Makran is that there is probably active deformation, not extensional. My feeling is that your normal faults correspond to minor expression of the system. For example, uplift would be more easily explained under compressional deformation rather than through normal faulting (part 4.3, p14.line 7). This would agree with what you state p15, l. 9-10: "Nevertheless, our results indicate that the internal part of the accretionary prism is still accommodating significant internal deformation and uplift, as also shown by Haghipour et al. (2012)". Also, the Jask terrace is nearby an active fault (Peyret et al., 2009). A map of active deformation is mandatory, with maybe the position of the trench (or the Makran frontal thrust). About MIS3 terraces. I am doubtful about your scenario (Beris-Pasbander area). OSL is not always perfect, and If in your scenario MIS3c corresponds to a true highstand, this is not the case for MIS3a. Considering that a large majority of the marine terraces documented on earth developed during a highstand, and that this

age leads to uplift rates largely higher than those shown by older terraces, I suspect a wrong scenario. Are you sure this does not corresponds to a MIS3c terrace?

Moderate comments. P.3 l.3-4. Uplift variations along strike is sometimes the consequence of the subduction of asperities/aseismic ridges (see the extensive literature on the subject). P3. Lines 16-24. There is probably along-strike variation in the convergence accommodated across the Makran, due at least to the East Lut fault (for example Walpersdorf et al., 2014). You must evoke this fact. P6 l14. "Platform carving usually happens at the beginning of the highstand, until the platform gets too wide and the wave energy too dissipated to carry out effective cliff erosion". I agree with this 'theoretical' statement. I say 'theoretical' because it has only been observed on numerical models. Actually, there is very few data about the nature, but cosmogenic nuclides contradict this view (Regard et al., 2012; Hurst et al., 2016). There is no consensus... P6 l17-19. Here again, this assertion is not always true. One counterexample may be found around the Somme Bay in France (50°10'N and 1°30'E). To the southwest (Ault) it passes to the Normandy chalk cliff coast. The area between Cayeux and Ault is made of sand ridges (former sand spit) sedimented over a rock shore platform. The sedimentation begun after the Holocene sea level rise at ∼5-6 kyrs BP, as observed on others lidos (like Venice). OSL Dating. I understand the technique but I am not a specialist. I have not reviewed this part. When there is no shoreline angle, you used the back of the terrace. I agree with you but I do not understand that you do not use it as a minimum value, instead of using it as an approximation? (p9, l20). P10 L15. You claim the radiocarbon are limited to 20ka. I do not agree! There are good calibration curves and also good radiocarbon results for more ancient periods. P11l2-5. You cannot write about Jask without mentioning the local active tectonics! P11 l25. I do not understand your argument. The lateral shift comes from the relief. Here, if the terraces are close, it indicates the relief is high enough to carve different terraces in a narrow zone. P14 l16-20. If terrace location and uplift are driven by the limits of earthquake rupture (Saillard et al., 2017), then, as earthquakes occur generally at the same depth on the subduction interface, the uplift distribution is expected to be function of the dis-
tance to the trench. Is it the case in your dataset, as suggested by the high uplift rates of the areas closest to the trench (Pasbander)? Supplementary material Figure C1 and tectonics. I do not find your diagnostic of normal motion is convincing. The Figure shows a fault that morphologically looks like a thrust, and you do not indicate what are the lithologies (and ages of them) affected by the fault.

Minor comments. TanDEM-X DEM (P5 l26). I used one in Peru and I observed a systematic offset. Have you verified your sea level corresponds to the elevation of 0? Digitization (p5 l27). What is the resolution to which you digitize? Normand et al. 2018. I had not been able to access to the dataset. Also, I am not sure it is at the right place in the reference list. Uncertainties p8 l23-27. Although it is clear in following parts in the manuscript, you might explain why you differentiate min/max where a correct formulation would be $\Delta U = \sqrt{(ãĂŰ((\Delta E + \Delta e)/(E-e))^2 + (\Delta A/A)ãĂŮ^2)}$ P9,l7. Due to the age uncertainties of the $\delta 18O$ records, the timing of the chosen sea-level curves is not well constrained (Shakun et al., 2015; Spratt and Lisiecki, 2016).My experience revealed that timing uncertainties are much less influencing the overall uncertainty than uncertainties on e. Sea also the interesting paper by Caputo (2007). P12 l24. I would not qualify MIS3 of lowstand. MIS2 and MIS4 are lowstands. Odd MIS must be highstands, even if MIS3 is a particular case. Figures: the graphs showing uplift rates and shoreline angle altitudes lack horizontal scale and cross section labels. Figure 3. They are not Uplift max and min but uplift rates max and min. Figure 4. "Pre-Holocene"

References Caputo, R.: Sea-level curves: Perplexities of an end-user in morphotectonic applications, Glob. Planet. Change, 57(3–4), 417–423, 2007. Hurst, M. D., Rood, D. H., Ellis, M. A., Anderson, R. S. and Dornbusch, U.: Recent acceleration in coastal cliff retreat rates on the south coast of Great Britain, Proc. Natl. Acad. Sci., 113(47), 13336–13341, doi:10.1073/pnas.1613044113, 2016. Peyret, M., Djamour, Y., Hessami, K., Regard, V., Bellier, O., Vernant, P., Daignières, M., Nankali, H., Van Gorp, S., Goudarzi, M., Chéry, J., Bayer, R. and Rigoulay, M.: Present-day strain distribution across the Minab-Zendan-Palami fault system from dense GPS transects, Geophys.

ESurfD
J. Int., 179(2), 751–762, doi:10.1111/j.1365-246X.2009.04321.x, 2009. Regard, V., Dewez, T., Bourlès, D. L., Anderson, R. S., Duperret, A., Costa, S., Leanni, L., Lasseur, E., Pedoja, K. and Maillet, G. M.: Late Holocene seacliff retreat recorded by 10Be profiles across a coastal platform: Theory and example from the English Channel, Quat. Geochronol., 11, 87–97, doi:10.1016/j.quageo.2012.02.027, 2012. Saillard, M., Audin, L., Rousset, B., Avouac, J.-P., Chlieh, M., Hall, S. R., Husson, L. and Farber, D. L.: From the seismic cycle to long-term deformation: linking seismic coupling and Quaternary coastal geomorphology along the Andean megathrust: Interseismic Coupling/Coastal Morphology, Tectonics, 36(2), 241–256, doi:10.1002/2016TC004156, 2017. Walpersdorf, A., Manighetti, I., Mousavi, Z., Tavakoli, F., Vergnolle, M., Jadidi, A., Hatzfeld, D., Aghamohammadi, A., Bigot, A., Djamour, Y., Nankali, H. and Sedighi, M.: Present-day kinematics and fault slip rates in eastern Iran, derived from 11 years of GPS data: Eastern Iran current deformation, J. Geophys. Res. Solid Earth, 119(2), 1359–1383, doi:10.1002/2013JB010620, 2014.

---

## Referee Comment (RC2) · Anonymous Referee #2 · 12 Dec 2018

An interesting paper looking at marine terraces along a subduction zone margin setting along the Iranian coast. The approach to use marine terraces to quantify spatial and temporal patterns of surface uplift is a standard approach for neotectonic studies worldwide. Thus, the paper presents a localised / regional case study at best. There is some attempt to step back and look at other subduction zone marine terraces worldwide but this doesn't happen until the end of the Discussion. I wonder if the paper would have more impact with this subduction zone context as the pervasive thread through the paper meaning the work would have more impact and a broader international appeal than is currently couched.

[Figure]

Some more technical points

1. DEM choice - TAN DEM data can be quite variable in quality since it is a new product. Some improved rationale as to why this dataset was used would be helpful. There are papers that test the visualisation and quantitative analysis of DEMs in geomorphology (Boulton and Stokes, 2018 Geomorphology) albeit for a fluvial audience.

2. Shoreline angles are critical for the uplift but the authors don't get to consider these until the methods. There should be marine terrace definitions and a clearer statement of shoreline angle significance earl;ier in the paper.

3. Shoreline angle erosion - there was no explanation as to how the shoreline angles were being eroded. Is it landslides and recessive cliff erosion by diffusion. needs clarification.

4. Mapping and stratigraphy - the paper text seems to be all about the dating when some much stronger text descriptions / statements abouty the field mapping, altitudes of the terraces, the relative stratigraphy of the terraces and the sediment sections are needed. The data is all in the (excelllent) diagrams and yet the text description of this essential framework is lacking from the main body of the paper. It's as though you are leaving the reader to look at the figures and to sort it out themself. Give the field framework a much stronger prevalence in the text.

5. Terrace tilting - this seems to be quite important for the uplift calculations and yet because the field descritpion is lacking the tilting of the terraces raises questions about the uplift rate calculation and broader interpretation. The titling (and the faults) need a stronger presence within the paper and some more careful explanation to demonstrate due care and attention.

6. Uplift rates - I couldn't get a sense of whether an uplift rate is high or low. My own published experience of marine terrace research from the Californian coast suggests to me that the Markan terraces here are giving very low uplift rate range values (min,

max and mean). So, could the authors just qualify what they mean by low or high and be careful when talking locally, regionally or globally.

---

## Referee Comment (RC3) · Jara-Muñoz (Referee) · 2 Jan 2019

This work studies the exceptional exposure of marine terraces along the Makran coast in Iran. This is a quite interesting study that attempt to integrate previous chronological constraints on these terraces with novel ages based on multiple approaches (OSL/14C/U/Th); in addition, the authors use detailed mapping and morphometry to estimate the patterns of surface deformation in this area, then used to discuss the source and mechanisms of such deformation in the context of the Makran subduction zone. The authors do a good work attempting to join the different ages obtained, which in some cases are not easy to interpret. One of the controversial points is the

presence of MIS 3 terraces, which are apparently related to localized high uplift rates. The presence of MIS 3 terraces is rare, but they discuss all the pros and cons for this interpretation. I personally find the paper clearly written with some minor typos and some issues; however, in general they clearly explain the logical steps behind their interpretations, which is the good way to do science (e.g. Section 5.2). The quality of the figures and the fashion used to display the distribution of the terraces are excellent and quite original, also the final interpretation about the possible mechanisms of tilting are nicely explained in the corresponding figure.

My main critics comes from:

1) The authors refer to active structures (Section 5.2 and 5.3.1) to explain local variations in uplift rates but the description or reference to these structures are poor, most of them based on speculative faults not observed on the field or mapped by other authors, then this part of the discussion becomes light and not convincing it all, specially about the kinematics of these structures. I must highlight that in general long wavelength deformation patterns are usually associated to deeper sources of deformation, such as the subduction megathrust, instead short wavelength deformation patterns are usually associated to shallower sources of deformation like crustal faults, I think that framing the interpretations based on these concepts may provide a more convincing discussion on the sources of deformation (e.g. asperities or subducted seamounts are related to deeper sources of deformation, instead abnormal local high uplift rates could be related to crustal faults, etc.., this would also help to complement section 5.3.2).

2) In sections 5.3.2 and 5.3.3 the authors discuss the deformation patterns and uplift rates of marine terraces in the context of the subduction earthquake cycle, I am aware that the historical records of earthquakes are scarce but I feel that the topics or paradigms mentioned in the introduction are weakly resolved, so I find their final interpretations and discussion a bit frustrating not fulfilling the expectation introduced at the beginning of the manuscript.

Minor/moderate comments:

Page 2 Line 28: "ten sequences"? Or ten levels of marine terraces?

Page 3 Line 27: "different tomographical properties"? this is ambiguous, what they describe in the tomography?

Page 4 Line 2: slope sedimentary rocks? Do you mean Sedimentary rocks of slope depositional environment?

Page 4 line 7: omega shaped?

Page 5, line 27: "TanDEM-X (0.4 arcsec/~12m . . .." This is repeated in page 2, there are also several other repetitions along the text

Page 7, line 13: "OSL dating", as I understood, you tried with quartz but then decided to use IRSL technique in feldspars, maybe is better state IRSL dating method instead of OSL, here and along the text as the results presented comes from IRSL.

Page 8, line 15: "nearest sea-level highstand" and also refer Jara-Muñoz et al., 2015. This is not correct it all, usually we use the age of the immediately preceding sea-level highstand, as the deposits are accumulated during the sea-level drop that follows the higstand.

Page 8, line 25: the author use equations to estimate the minimum and maximum uplift rate, why do not propagate the errors like have been classically done by other authors before? (e.g Gallen et al., 2014)

Page 9, line 18: The concept of eroded shoreline angle sounds weird, and also the way to estimate its elevation, is not clear which part is used to estimate the elevation of this feature, as the paleo-platform can extend seawards for long distances its elevation can display wide variations.

Page 9, line 21: "their altitude might underestimate the reality"? do you mean their altitude might represent minimum estimate?

Page 9, line 24: "... we calculated uplift rates for each sample..." but before you mention that uplift rates are not calculated using the sample elevation but the shoreline angles, this is contradictory.

Page 9, line 25: this is contradictory, here it says "... we subtracted the sediment thickness observed on the field..." and in line 29 it says "... we subtracted a general value for the thickness of the sediments..." so, what you really did?

Page 10, line 11: "results of radiometric....." this definitely is not the best way to start a paragraph, please use topic sentences here and in some other paragraphs of the manuscript. Page 11, line 26: remove "mostly"

Page 14, line 5: "The observation of normal faulting in a convergence context is intriguing". Actually this is very common in the fore arc of the Andes where normal fault are a result of crustal bending and several other processes, not suggesting subsidence (e.g. Lowless et al., 2010; Melnick et al., 2012; Melnick et al., 2009).

Page 15, line 9: "Those few terraces that are not tilted. ... Might provide insights on the uplift component directly linked to subduction dynamics. ..." I disagree, as mentioned in the major comments is the pattern of deformation which may provide insight about the mechanisms of deformation either long or short wavelength may provide insights about deeper or shallower sources related to crustal structures or megathrust deformation.

Finally, I must say that this is an interesting paper with high potential to be a great contribution after improving and correcting some of the issues described before.

References:

Gallen, S., Wegmann, K., Bohnenstiehl, D., Pazzaglia, F., Brandon, M., Fassoulas, C., 2014. Active simultaneous uplift and margin-normal extension in a forearc high, Crete, Greece. Earth Planet. Sci. Lett. 398, 11e24.

Loveless, J.P., Allmendinger, R.W., Pritchard, M.E., Gonzalez, G., 2010. Normal and reverse faulting driven by the subduction zone earthquake cycle in the north- ern

Chilean fore arc. Tectonics 29.

Melnick, D., Moreno, M., Motagh, M., Cisternas, M., Wesson, R.L., 2012. Splay fault slip during the Mw 8.8 2010 Maule Chile earthquake. Geology 40, 251e254.

Melnick, D., Bookhagen, B., Strecker, M.R., Echtler, H.P., 2009. Segmentation of megathrust rupture zones from fore-arc deformation patterns over hundreds to millions of years, Arauco peninsula, Chile. J. Geophys. Res. Solid Earth 114, B01407.

---

## Author Comment (AC1) · 29 Jan 2019

Pdf + supplementary material in zip file below.

Please also note the supplement to this comment:
https://www.earth-surf-dynam-discuss.net/esurf-2018-78/esurf-2018-78-AC1-supplement.zip

---

## Referee Report (RR1)

Dear Professor Mudd

Thanks for the opportunity to review this paper. The author did a good job addressing all my previous comments and corrections. One of the major issues, the emphasis of the paper is now equilibrated; intro and discussion are matching pretty well and the focus was switched from seismogenesis to mechanisms of surface deformation. Deformation mechanisms are now discussed considering the wavelength of structures and patterns of surface deformation, which is a major improvement and fix one of the main problems highlighted in the previous versions. Minor problems like typos and repetitions have been corrected and the text considerable improved. I am really satisfied with this new version and I think is ready for publication.

All the best

Dr. Julius Jara
Universität Potsdam

**Response to RC3:**

*This work studies the exceptional exposure of marine terraces along the Makran coast in Iran. This is a quite interesting study that attempt to integrate previous chronological constraints on these terraces with novel ages*
*based on multiple approaches (OSL/14C/U/Th); in addition, the authors use detailed mapping and morphometry to estimate the patterns of surface deformation in this area, then used to discuss the source and mechanisms of such deformation in the context of the Makran subduction zone. The authors do a good work attempting to join the different ages obtained, which in some cases are not easy to interpret. One of the controversial points is the presence of MIS 3 terraces, which are apparently related to localized high uplift rates. The presence of MIS 3*
*terraces is rare, but they discuss all the pros and cons for this interpretation. I personally find the paper clearly written with some minor typos and some issues; however, in general they clearly explain the logical steps behind their interpretations, which is the good way to do science (e.g. Section 5.2). The quality of the figures and the fashion used to display the distribution of the terraces are excellent and quite original, also the final interpretation about the possible mechanisms of tilting are nicely explained in the corresponding figure.*

*My main critics comes from:*
*1) The authors refer to active structures (Section 5.2 and 5.3.1) to explain local variations in uplift rates but the description or reference to these structures are poor, most of them based on speculative faults not observed on the field or mapped by other authors, then this part of the discussion becomes light and not convincing it all,*
*specially about the kinematics of these structures. I must highlight that in general long wavelength deformation patterns are usually associated to deeper sources of deformation, such as the subduction megathrust, instead short wavelength deformation patterns are usually associated to shallower sources of deformation like crustal faults, I think that framing the interpretations based on these concepts may provide a more convincing discussion on the sources of deformation (e.g. asperities or subducted seamounts are related to deeper sources*
*of deformation, instead abnormal local high uplift rates could be related to crustal faults, etc.., this would also help to complement section 5.3.2).*
First part. We agree that speculating on the presence of normal faults is an easy an unconvincing interpretation, and so, we have entirely removed it from the text.
We thank you for your valuable input regarding the interpretation of uplift patterns. We have taken your
comments and suggestion into account and have thoroughly modified section 5.3 accordingly.

This solve the critic 1, nice addition

*2) In sections 5.3.2 and 5.3.3 the authors discuss the deformation patterns and uplift rates of marine terraces in the context of the subduction earthquake cycle, I am aware that the historical records of earthquakes are scarce but I feel that the topics or paradigms mentioned in the introduction are weakly resolved, so I find their final*
*interpretations and discussion a bit frustrating not fulfilling the expectation introduced at the beginning of the manuscript.*
We have modified the text to make it clearer that uplift rates patterns obtained from marine terraces are averaged over multiple seismic cycles and therefore, only give relevant information on the long-term signals rather than on the scale of individual earthquakes. This was adapted in the introduction (p.2 l.5).

The critic 2 was correctly addressed

*Minor/moderate comments:*

*Page 2 Line 28: "ten sequences"? Or ten levels of marine terraces?*

We are not referring to terrace levels, but to 10 different terrace sequences (1 in Jask, 1 in Tang, 1 in Gurdim, 2 in Konarak, 2 in Chabahar-Ramin, 1 in Lipar, 2 in Pasabander, as shown by the N-S profile in the terrace maps), each with its own different succession of levels. In fact, we really end up looking at the global lateral variation rather than punctual profiles, but these profiles represent the different "independent" sequences observed.

Thanks for this explanation, the meaning of sequence was confusing in the earlier version

*Page 3 Line 27: "different tomographical properties"? this is ambiguous, what they describe in the*
*tomography?*
We have edited the text to make it clearer that different seismic wave velocities between the two segments have been detected with tomography (p.3 l.29).

Well addressed

*Page 4 Line 2: slope sedimentary rocks? Do you mean Sedimentary rocks of slope depositional environment?*
Yes, we have edited the text to make it clear (p.3 l.3).

Correction accepted

*Page 4 line 7: omega shaped?*
It refers to the geographical shape of the bay similar to the greek letter omega $\Omega$ (a term often used to describe the Makran bays) (see fig. 1).

Thanks for this explanation

*Page 5, line 27: "TanDEM-X (0.4 arcsec/_12m . . .." This is repeated in page 2, there are also several other repetitions along the text*
We have edited the text to correct those mistakes as much as possible.

Yes, I notice the current version much more improved

*Page 7, line 13: "OSL dating", as I understood, you tried with quartz but then decided to use IRSL technique in feldspars, maybe is better state IRSL dating method instead of OSL, here and along the text as the results presented comes from IRSL.*
IRSL is one of the methods of optically stimulated luminescence dating; hence, the terms are synonyms. We use the term OSL, because it is more commonly used and recognizable by the scientific community, but we state in
the methods that we used IRSL (edited and emphasized, p.7 l. 23).

Not addressed.

Maybe thermo luminescence would be a more general term, but OSL or also know as "quartz OSL" is a method specifically for the analysis of quartz using the full light spectra. IRSL or post-IRSL are not a type of OSL.

*Page 8, line 15: "nearest sea-level highstand" and also refer Jara-Muñoz et al., 2015. This is not correct it all, usually we use the age of the immediately preceding sea-level highstand, as the deposits are accumulated during the sea-level drop that follows the higstand.*
We apologize for this mistake. We have corrected the text (p.8 l.18).

No need to apologize, nice additions in text

*Page 8, line 25: the author use equations to estimate the minimum and maximum uplift rate, why do not propagate the errors like have been classically done by other authors before? (e.g Gallen et al., 2014)*
We use the same method as in Pedoja et al. (2018a, 2018b). This method slightly overestimate the error
compared to the standard error propagation. We have edited the text (p.8 l.32) and added a tab in the supplementary data B (B.4) where we calculate uplift rates and errors using the standard error propagation. The uplift rates profiles from the map figures was left the same.

This is well addressed

*Page 9, line 18: The concept of eroded shoreline angle sounds weird, and also the way to estimate its elevation, is not clear which part is used to estimate the elevation of this feature, as the paleo-platform can extend seawards for long distances its elevation can display wide variations.*

We are aware that it is an uncommon (unique?) feature, and it is not straightforward to describe it. We have modified Fig. 2a and emphasized the reference to field pictures (Fig 2h, and Figs H in dataset) that illustrate the concept (p.9 l.23).

This correction was correctly addressed, but I never heard about a terrace feature like this before, would be nice to see it on the field.

*Page 9, line 21: "their altitude might underestimate the reality"? do you mean their altitude might represent minimum estimate?*

Yes. We have edited the text to make that clearer (p.9 l.27).

Correction accepted

*Page 9, line 24: ". . . we calculated uplift rates for each sample. . ." but before you mention that uplift rates are not calculated using the sample elevation but the shoreline angles, this is contradictory.*

We have changed the text from: (p.9 l.29)

We calculated uplift rates for each sample using the terrace shoreline angle situated directly northwards of the sample (Table 3) (i.e., perpendicular to the trench) …

To

We calculated uplift rates at the longitude of each sample using the terrace shoreline angle…

Ok, this more clarifying and corrections are well addressed

*Page 9, line 25: this is contradictory, here it says ". . . we subtracted the sediment thickness observed on the field. . ." and in line 29 it says ". . . we subtracted a general value for the thickness of the sediments..." so, what*

*you really did?*

We did both. For the samples, we were able to visit the shoreline angle on the field and measure the exact thickness of the sediments. However, for the large scale calculation that we do for the whole ranges of shoreline angles extracted from the DEM (>>100 km), we only use a general value because we could not physically go and measure the sediment thicknesses everywhere. We have adapted the text to make that clear (p.9 l.30-32).

Ok, this more clarifying and corrections are well addressed

*Page 10, line 11: "results of radiometric. . ..." this definitely is not the best way to start a paragraph, please use topic sentences here and in some other paragraphs of the manuscript. Page 11, line 26: remove "mostly"*

We have edited the text accordingly.

Addressed

*Page 14, line 5: "The observation of normal faulting in a convergence context is intriguing". Actually this is very common in the fore arc of the Andes where normal fault are a result of crustal bending and several other processes, not suggesting subsidence (e.g. Lowless et al., 2010; Melnick et al., 2012; Melnick et al., 2009).*

We agree that normal faults actually are observed in many subduction zones (not only in Chile, which is the typical example, but also in Japan, NZ, Greece,…), it nonetheless remains an intriguing fact not always correctly understood. We have adapted this part in section 5.3.

Addressed

*Page 15, line 9: "Those few terraces that are not tilted. . .. Might provide insights on the uplift component directly linked to subduction dynamics. . .." I disagree, as mentioned in the major comments is the pattern of deformation which may provide insight about the mechanisms of deformation either long or short wavelength may provide insights about deeper or shallower sources related to crustal structures or megathrust deformation.*

We agree, see comments to the first part.

Yes, I notice that your discussion is now more solid, good work!!

*Finally, I must say that this is an interesting paper with high potential to be a great contribution after improving and correcting some of the issues described before.*